# Magnetoactive bistable soft actuators for programmable large shape transformations at low magnetic fields

Hao Wen[1,2], Zihao Shao[1,2], Yuxuan Sun [1,2], Chiyuan Ma[1,2], Feihong Xiang[1,2], Liangyu Xia[1,2], Xinhui Zhu[1,2], Xiaoxiang Li[1,2], Liang Li[1,2] ✉ & Quanliang Cao [1,2] ✉

As the demand for advanced actuation strategies in soft robotics and intelligent material systems grows, magnetoactive soft actuators have attracted increasing attention for their ability to achieve flexible shape transformations through remote and untethered control. However, existing designs typically rely on continuous high magnetic fields to generate large deformations, limiting both efficiency and applicability, especially under constrained boundary conditions. Here we report a hemispherical bistable soft actuator embedded with magnetic microparticles, which enables substantial shape changes under low-intensity pulsed magnetic torques and remains stable in two configurations without external fields. We analyze the relationship between design parameters and actuator performance to clarify the bistable mechanism, and show that the actuator can achieve a large shape change ratio exceeding 0.8 under magnetic fields below 20 mT. We further demonstrate its versatility through three applications: a high-efficiency soft pump with closed-loop fluid control, a reprogrammable metamaterial, and a variable-stiffness soft gripper.

Soft active materials exhibit broad scientific importance and application potential due to their distinctive capabilities in shape transformation and active control. These materials can undergo complex shape changes response to external stimuli such as light[1,2], heat[3,4], and magnetic fields[5–7], enabling them to effectively emulate the dynamic motion and adaptive behaviors observed in biological and engineered systems. Among these stimuli, magnetic fields offer a practical and effective method for actuation, as they allow magnetic soft actuators to achieve non-contact operation, high controllability, and excellent penetration capabilities[8]. These advantages make magnetic soft actuators, especially the recently developed actuators based on magnetic torque and hard magnetic composites, well suited for applications requiring multimodal deformation and locomotion as well as minimal invasiveness, such as in biomedical[9–12] and biomimetic robotics[13–15]. However, current magnetoactive soft actuators primarily rely on the deformation principle involving a competition between external magnetic torques and internal elastic forces[16,17], which

constrains their capacity for large-scale shape manipulation under low magnetic fields and limits shape stability without continuous magnetic actuation. Overcoming this limitation is pivotal for expanding the range of applications, particularly those requiring precise magnetic field control with electromagnetic coils. This is because excessive magnetic fields and continuous magnetic output necessitate higher electrical currents and extended activation times, inevitably leading to inefficiency and shortened lifespan of coils[18,19].

To address the aforementioned limitations, integrating bistable structures into magnetically actuated systems may provide a promising breakthrough. Bistability, which refers to the ability of a structure to maintain two stable states after experiencing mechanical instability, has emerged as a powerful strategy to enhance the performance and functionality of soft actuators[20,21]. By utilizing bistable mechanisms, it is possible to achieve force amplification, high-speed movements, and shape retention without continuous-applied external actuation[22,23]. When integrated with magnetic materials and corresponding magnetic

[1]Wuhan National High Magnetic Field Center, Huazhong University of Science and Technology, Wuhan, China. [2]School of Electrical and Electronic Engineering, Huazhong University of Science and Technology, Wuhan, China. ✉e-mail: Liangli44@hust.edu.cn; quanliangcao@hust.edu.cn

actuation strategies, these bistable elements offer a synergistic combination of contactless actuation and mechanical stability, further improving system reliability, energy efficiency, and controllability. An example is provided by Tang et al.[24], who utilized the rapid response and high energy release of a bistable mechanism to develop a pyramid-shaped, magnetically actuated jumping robot. This system demonstrated millisecond-scale actuation and achieved a self-propulsion height that exceeded 100 times its own body length, which highlights its high energy conversion efficiency under low magnetic fields. This emerging field has seen increasing research activity in recent years, resulting in the creation of diverse magnetoactive bistable structures, including beam-shaped[25–27], dome-shaped[28–31], origami and kirigami structures[24,32–34], spanning from one-dimensional (1D) to three-dimensional (3D) configurations. Among them, dome-shaped configurations, as a representative 3D bistable architecture, are promising for applications in magnetoactive soft actuators, owing to their structural simplicity and their ability to achieve more substantial volumetric transformations than planar structures[35]. However, despite their structural advantages, current research on dome-shaped magnetoactive bistable soft actuators has paid limited attention to improving their shape-switching performance. In particular, under practical conditions, involving fully edge-constrained conditions, their deformation capability remains highly restricted, hindering the full exploitation of the bistable mechanism. As a result, achieving large-scale shape transformations under low magnetic fields ($B < 100$ mT),

specifically a shape change ratio exceeding 0.5, where the ratio refers to the actuator's maximum deformation divided by a characteristic structural dimension (e.g., diameter or length), remains a major challenge (Supplementary Information, Note S1, and Table S1). This limitation constrains the broader deployment and functional advancement of magnetoactive soft actuators in complex real-world applications.

In this work, inspired by the bistable properties of contact lenses (Fig. 1a), we developed a magnetoactive soft actuator based on a hemispherical structure belonging to the dome-shaped category with bistable functionality, which is capable of achieving large shape transformations under low magnetic fields and edge constraints. As schematically illustrated in Fig. 1b, the soft actuator exhibits two distinct stable states, convex (State A) and concave (State B), in the absence of external loads, and it can switch between these two stable states by adjusting the direction of an externally applied magnetic field. The shape transformation is driven by the torque induced by the misalignment between the magnetization of the hemispherical shell and the applied external magnetic field. Prior to the transformation, the applied magnetic field is aligned antiparallel (180°) to the magnetization direction, theoretically resulting in zero initial torque[36]. However, even a slight perturbation—such as a small deviation in the local magnetization direction—can generate local magnetic torques[37], amplifying non-zero torque regions and triggering a shape transition between the convex and concave states when the magnetic field

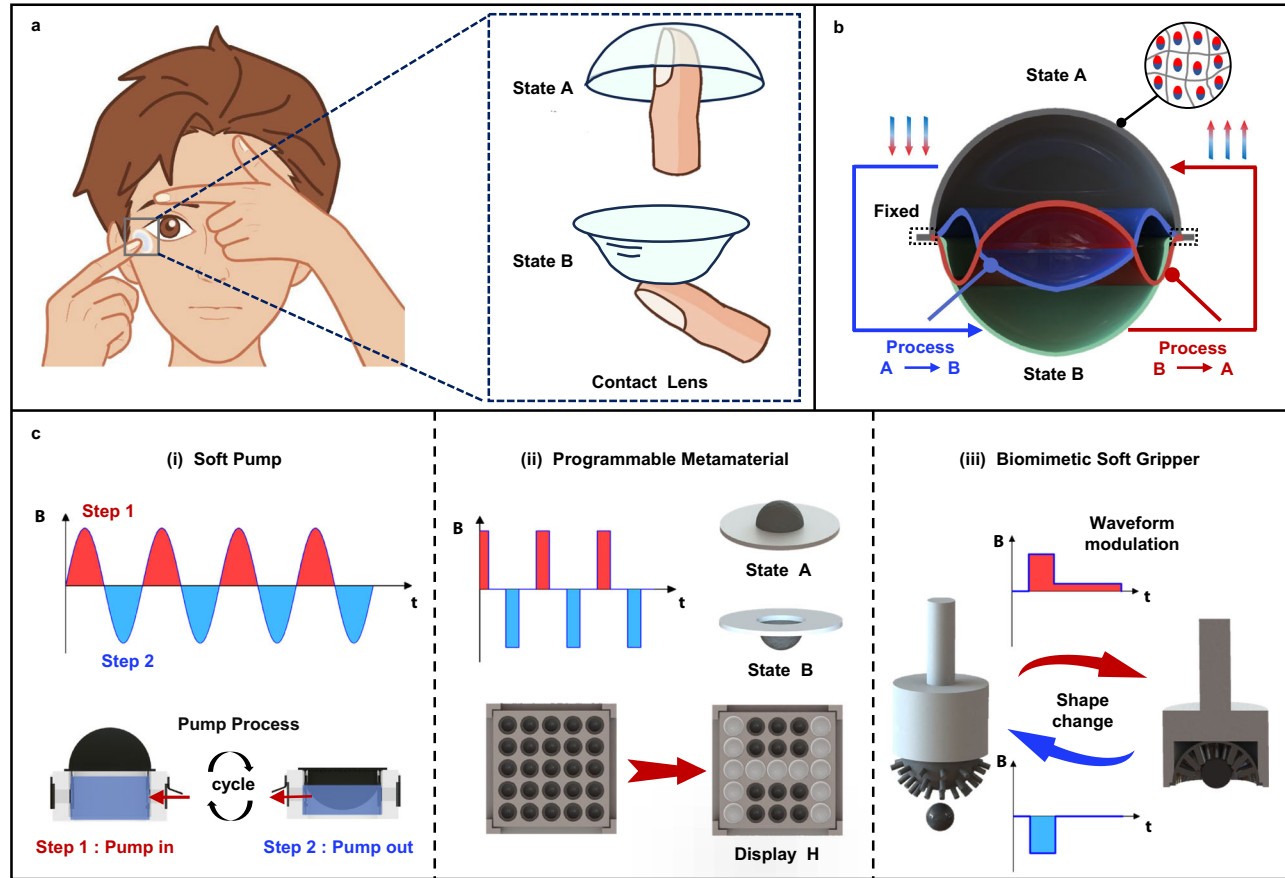

**Fig. 1 | Structural principle and functional applications of the proposed magnetoactive bistable hemispherical soft actuators. a** Inspired by the structure of contact lenses, the actuator unit exhibits a bistable structural principle. **b** The actuation principle of the actuator unit involves axially magnetized hard magnetic particles, which generate magnetic torque under an applied magnetic field, thereby causing state switching. **c** The soft actuators demonstrate multifunctionality in the

following applications: (i) Magnetoactive soft pumps that enable efficient gas and fluid pumping under alternating magnetic fields. (ii) Programmable metamaterials with shape configurations controlled by the direction of a pulsed magnetic field. (iii) Biomimetic sea anemone soft grippers that perform grasp-release actions through magnetic field direction control, with stiffness adjustments made via magnetic field waveform modulation.

exceeds a critical value. Unlike existing methods that rely on self-buckling[28] or externally induced pre-stress[31] to form dome-shaped bistable structures from flat sheets, our approach directly integrates a 3D hemispherical shell during the fabrication process. This approach enhances the spatial changes associated with bistable switching, thereby facilitating large shape transformations. Notably, our hemispherical shell integrates a soft composite material composed of Neodymium-iron-boron (NdFeB) magnetic microparticles uniformly dispersed within a silicone matrix. These microparticles are magnetized by an axial pulsed high magnetic field, imparting permanent magnetic properties to the actuator. In contrast to dome-shaped magnetic bistable soft actuators that utilize gradient magnetic forces[29,30], our developed actuator derives its deformation mechanism from magnetic torque, independent of the magnetic field gradient. This is advantageous for remote control, as force fields tend to decay more rapidly in magnetic gradient force loading modes. Meanwhile, the externally applied magnetic torque acts as a transient disturbance rather than directly causing large deformation, thus overcoming the typical high and continuous magnetic torque demands of conventional magnetoactive soft actuators for large deformation. Collectively, these mechanisms enable our actuator to achieve large shape transformations under low magnetic field conditions.

Our magnetoactive bistable soft actuator can operate independently and integrate well with other components for versatile applications (Fig. 1c). For instance, leveraging its capability for large shape transformations under low magnetic fields, we have developed a magnetoactive soft pump under alternating magnetic fields (Fig. 1c (i)). Additionally, utilizing the bistable characteristics of the actuator, we created a programmable metamaterial with a soft actuator array capable of multiple stable states, enabling magnetic reprogramming and autonomous display functionalities (Fig. 1c (ii)). Moreover, we engineered a biomimetic, sea anemone-inspired robotic gripper based on the actuator structure, which demonstrates rapid responsiveness to stimuli, maintains a secure grip without the need for continuous magnetic fields, and adjusts its stiffness to handle heavy objects (Fig. 1c (iii)). The implementation of these magnetoactive soft actuator applications will effectively demonstrate the advantages of combining magnetic actuation strategies with a hemispherical bistable structure, which provides a useful reference for broader applications.

## Results
### Design and performance of magnetoactive bistable soft actuator

A straightforward method was employed for the fabrication of magnetoactive bistable soft actuators, which involves the molding of a magnetic composite of Ecoflex 00-20 silicone elastomer and NdFeB microparticles, using a 3D-printed hemispherical mold, as shown in Fig. 2a. Building on this, we explored the relationship between the bistable properties of the hemispherical shell and its structural parameters by fabricating actuator samples with different curvature radii ($R$) and thicknesses ($h$). As shown in Fig. 2b, the hemispherical soft actuators can exhibit either bistable or non-bistable (monostable) characteristics under different structural parameters. Specifically, as the ratio of the radius to thickness increases, bistability becomes more likely, which is consistent with the bistable boundary model presented in Supplementary Information, Note S2. As an example, the deformation processes of monostable and bistable shells are provided in Supplementary Information, Fig. S2, and Supplementary Movie 1. Under bistable conditions, both concave-up and convex-down shapes can maintain their stability in the absence of an external magnetic field, as shown in Supplementary Information, Figs. S3-S4, and Supplementary Movie 2. We also studied the influence of substrate materials on magnetic field-induced bistable switching by measuring Z-axis compression resistance of polydimethylsiloxane (PDMS) and Ecoflex 00-20 silicone-based actuators (Fig. 2c (i)), using a permanent magnet (N35,

$D$50 mm × $H$20 mm, $D$ for diameter, $H$ for height) to control the applied magnetic field. The compression resistance of PDMS-based actuators is more than 50 times higher than that of silicone-based actuators, due to the higher Young's modulus of PDMS. This explains why, as shown in Fig. 2c (ii), although PDMS-based actuators exhibit bistable characteristics (states A and B), they struggle to switch states under the applied magnetic field. In contrast, silicone-based actuators can easily achieve state switching under the same magnetic conditions. Therefore, we selected Ecoflex 00-20 silicone as the actuator material in this study and further investigated the magnetic field required for shape transformation, referred to as the switching magnetic field, in silicone-based actuators ($R = 10$ mm, $h = 0.5$ mm) with different mass fractions of magnetic powder (Fig. 2d). The magnetic powder content influences the switching magnetic field by affecting the Young's modulus and residual magnetization strength. Experimental results show that, in the given tests, the minimum switching magnetic field is 20 mT when the magnetic powder content is 50 wt%. Thus we selected a combination of 50 wt% magnetic powder and silicon substrate material for subsequent experiments. Additionally, we tested the variation in the switching magnetic field with different shell thicknesses and radii. As shown in Fig. 2e, the switching magnetic field increases with shell thickness but decreases with shell radius. In our tests, the actuator with the largest radius ($R = 20$ mm) and thinnest shell ($h = 0.5$ mm) requires only about 10 mT to switch, enabling large shape transformations under this low field. Similar to the effect induced by increasing the Young's modulus, a greater thickness substantially enhances the bending stiffness, thereby increasing the difficulty of deformation and the magnitude of the required switching magnetic field. In contrast, the mechanism underlying the influence of shell radius is more complex. Although the bending stiffness is theoretically independent of the shell radius, variations in radius lead to observable changes in shell volume, edge constraint conditions, and local strain distribution during deformation. Consequently, the observed increase in the switching magnetic field with decreasing shell radius could result from the combined effect of these multiple interacting factors.

We then explored the magnetic response and shape transformation of our magnetoactive bistable soft actuators, comparing them with conventional circular magnetoactive actuators. For the comparison, both the bistable actuator and the conventional circular actuator are made from a magnetic soft composite material containing 50 wt% NdFeB, with identical planar geometric parameters (radius $R = 20$ mm, height $h = 1$ mm). The bistable actuator is magnetized axially, while the circular actuator undergoes pre-deformation using a convex mold with a height equal to its radius $R$ before being magnetized axially (see Supplementary Information, Fig. S5 for detailed geometry). Upon exposing both actuators to external magnetic fields (Fig. 3a–c), we observe that, compared to the conventional circular actuator, the bistable actuator exhibits distinct deformation characteristics and demonstrates superior deformation capabilities at low magnetic fields. Specifically, unlike the circular actuator, which shows a gradual increase in deformation with increasing magnetic field, the bistable actuator displays a pronounced discontinuous transition around 40 mT, with minimal shape change before and after this threshold (Fig. 3a). This bistable behavior is exhibited as a sudden deformation, either from convex to concave or vice versa, showing a certain degree of symmetry (Fig. 3b). Notably, this abrupt deformation mode allows the bistable actuator to achieve a much larger shape change at lower magnetic fields. For example, at 40 mT, the bistable actuator reaches a positive height exceeding 19 mm, while the conventional circular magnetoactive actuator deforms by less than 5 mm (Fig. 3c). Even when the magnetic field is increased by an order of magnitude, the deformation of the conventional actuator still fails to match that of the bistable actuator. This phenomenon occurs because, in our bistable actuator, the magnetic field primarily overcomes the resistance to state transitions to achieve large displacements, unlike the

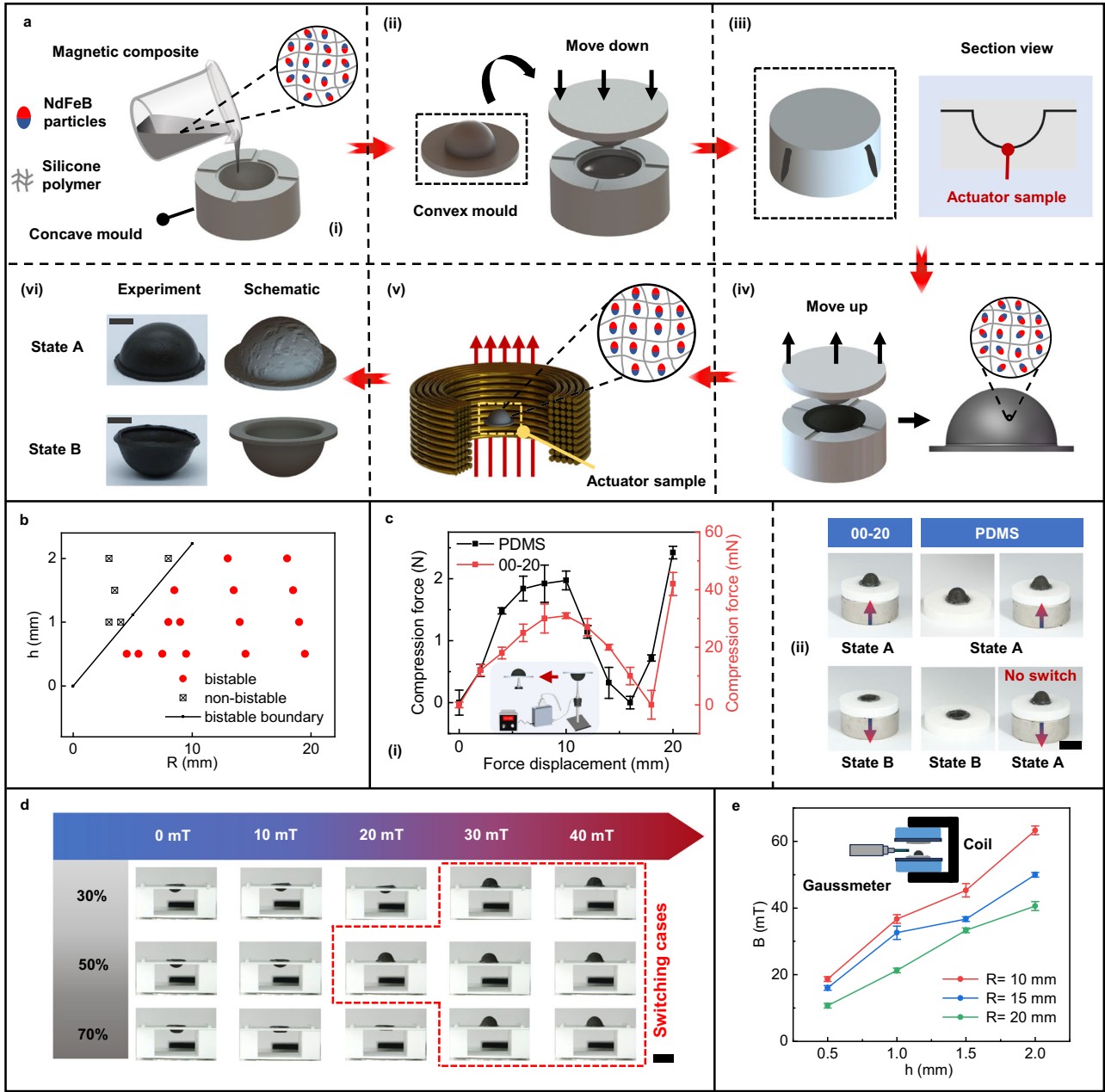

**Fig. 2 | Preparation process and bistable behavior of the proposed magnetoactive bistable hemispherical soft actuators. a** Preparation process of the actuators. (i) Pouring the NdFeB/silicone mixture into the concave spherical shell mold. (ii) and (iii) Vertically pressing the convex mold until the mixture overflows from the edges of the groove. (iv) Demolding the convex mold to obtain the actuator sample from the concave mold. (v) Axial magnetization treatment of the actuator sample under a 3 T pulsed magnetic field. (vi) Experimental and schematic diagrams of the actuator in two stable states. Scale bar: 10 mm. **b** Bistable characteristics of actuator samples with different curvature radii and thicknesses under an external load. **c** Deformation resistance and process of actuator samples with different soft substrate bases during deformation. (i) Variation curves of deformation resistance during the deformation process for PDMS and silicone Ecoflex 00-20 based actuators ($n = 3$, data are presented as mean values +/− SD). The Fig. 2c

was partly generated using Servier Medical Art, provided by Servier, licensed under a Creative Commons Attribution 3.0 unported license[55]. (ii) Deformation process comparison for two different actuator bases, where the silicone Ecoflex 00-20 base enables magnetic switching between two states under the influence of a permanent magnet, while the PDMS base, despite having two stable states, cannot achieve magnetic switching under a permanent magnet (permanent magnet diameter $D = 50$ mm, thickness $H = 20$ mm, surface magnetic field strength 367 mT, actuator sample size $R = 10$ mm, $h = 1$ mm). Scale bar: 20 mm. **d** Experiment on the switching magnetic field of actuators with varying magnetic powder content (actuator dimensions $R = 10$ mm, $h = 0.5$ mm). Scale bar: 20 mm. **e** Experiment on the switching magnetic field of actuators with different radii and thicknesses ($n = 3$, data are presented as mean values +/− SD). Source data are provided as a Source Data file.

conventional circular actuator, which requires excessive deformation. This results in effective performance even at lower field strengths.

To further evaluate the stability and repeatability of the shape transformation process under continuous loading, the soft actuator was subjected to a square wave pulsed magnetic field (amplitude $B = 50$ mT, period $T = 2.5$ s, $B$ for magnetic field, $T$ for time) for 100

cycles. Measurements taken with a laser displacement sensor confirmed that the actuator's shape transformation remained highly stable and repeatable throughout multiple cycles (Fig. 3d and Supplementary Movie 3), with no visible surface creases, indicating robust durability. Furthermore, to evaluate the impact of the magnetic field on the configuration transition process, we applied direct current (DC) and

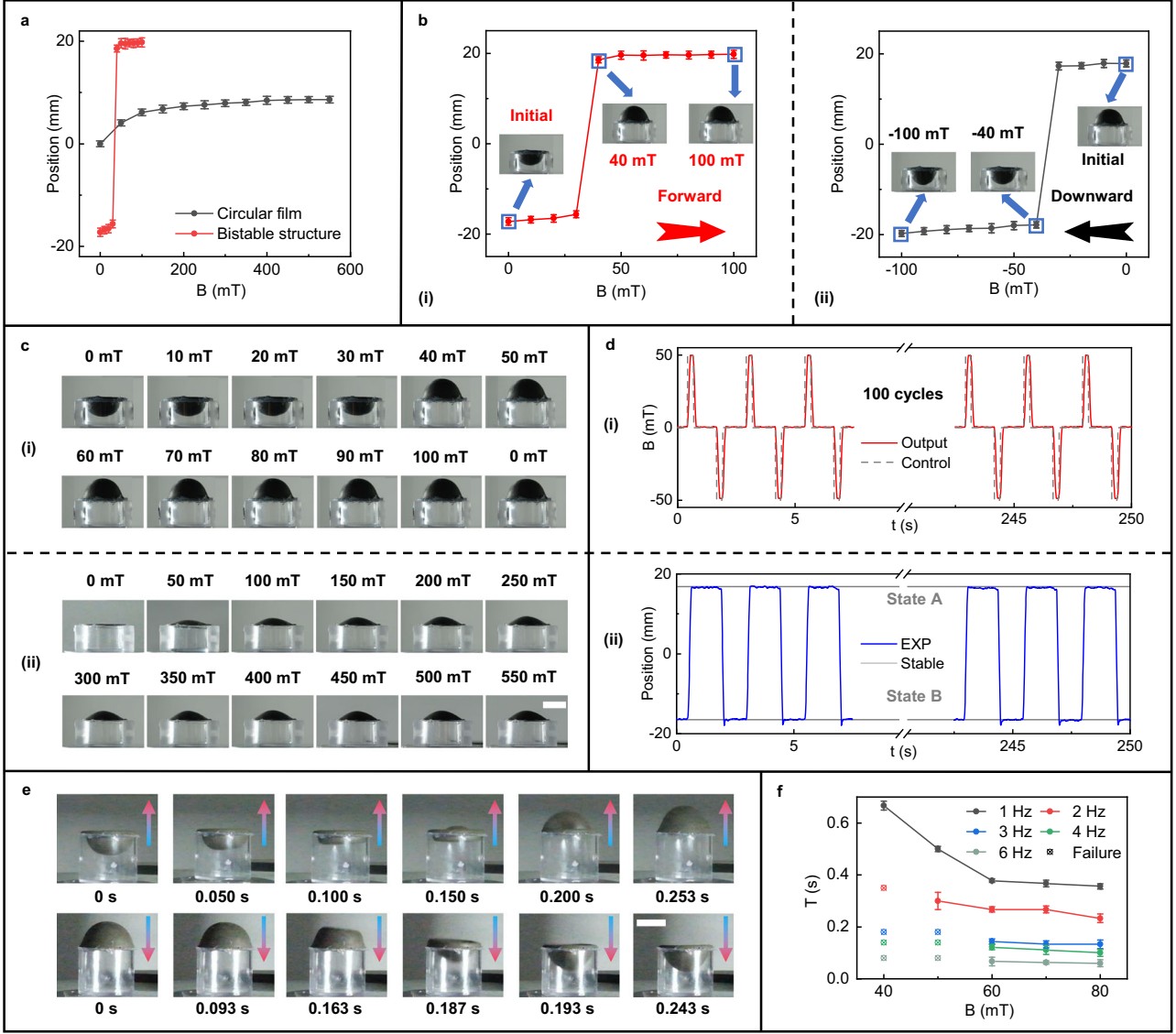

**Fig. 3 | Deformation and shape transformation of the proposed magnetoactive bistable hemispherical soft actuators. a** Deformation (i.e., change in the central point position) of our hemispherical actuator samples and conventional circular actuator samples under different magnetic fields ($n = 3$, data are presented as mean values ±SD). **b** Deformation curves of the hemispherical samples under positive (i) and negative (ii) magnetic fields, showing symmetric switching magnetic fields for both processes ($n = 3$, data are presented as mean values ±SD). **c** Images of deformation results of the hemispherical samples (i) and circular samples (ii) under magnetic fields. Scale bar: 20 mm. **d** Reversible bistable switching of our hemispherical soft actuator under a continuous pulsed magnetic field for 100 cycles ($T = 2.5$ s, $B = 50$ mT): time curve of the applied magnetic field (i) and the corresponding actuator displacement (ii). EXP: Experimental data. **e** Experimental images of the hemispherical sample switching between two stable states under a 60 mT DC magnetic field. Scale bar: 20 mm. **f** Switching time curve of the hemispherical sample under AC magnetic fields with different frequencies and intensities ($n = 3$, data are presented as mean values ±SD). Source data are provided as a Source Data file. Geometric dimensions of the hemispherical samples and circular samples are radius $R = 20$ mm and thickness $h = 1$ mm.

alternating current (AC) magnetic fields with varying amplitudes and frequencies. Under a 60 mT DC field, the bistable actuator exhibited forward and reverse switching times of 0.243 and 0.253 s, respectively (Fig. 3e, Supplementary Movie 4). Under an AC field, we measured and analyzed how the switching time varied with different magnetic field parameters (Fig. 3f and Supplementary Movie 5). At a fixed frequency, increasing magnetic field strength reduces the switching time. For instance, at 1 Hz, increasing the field strength from 40–80 mT reduces the switching time from approximately 0.667–0.356 s. Similarly, at a fixed field strength, increasing frequency shortens the switching time. For example, at 60 mT, the switching time decreases from approximately 0.378–0.067 s as the frequency increases from 1–6 Hz. Additionally, switching stability depends on both the frequency and strength of the magnetic field. For example, a 50 mT field allows stable switching up to 2 Hz but fails above 3 Hz. Therefore, in practical applications, the switching characteristics of the actuator can be adjusted and improved by synergistically controlling the amplitude and frequency of the magnetic field.

## Magnetoactive soft pumps

Soft pumps offer practical advantages in applications such as manipulation, locomotion, and wearable technologies due to their low cost, compliance, and multi-degree-of-freedom capabilities[38,39]. However, traditional soft pumps often rely on bulky air compression systems, which constrain their use in mobile and wearable devices. Recent advancements have shifted focus towards soft pumps driven by alternative sources such as electricity[40,41], heat[42,43], and magnetic fields[15,44–46], expanding their potential applications. In this work, we

leveraged the advantages of the developed magnetoactive bistable soft actuators, which enable substantial morphological changes under low magnetic fields. We have applied these actuators to the development of a magnetoactive soft pump and demonstrate its effectiveness in both gas and liquid delivery applications.

Taking pumping to the right as an example, the magnetoactive soft pump, which comprises a magnetoactive bistable soft actuator and two sets of magnetic soft valves, operates by exploiting the differential deformation responses of these components under alternating magnetic fields. As shown in Fig. 4a, under the influence of an alternating magnetic field, the shape change of the soft actuator leads to continuous changes in the chamber volume, while the soft valves on both sides alternately open and close, thereby achieving the pumping action. The composition of this pump and pumping principle are similar to the soft pump system with a folded magnetic diaphragm proposed by Lin et al.[15], but there are differences in the structure of the magnetic actuator, the magnetic-induced deformation mechanism, and the soft valve design. In this application, the pump's valves are composed of magnetic soft materials embedded with NdFeB particles, and the magnetization design of the left and right valves is shown in Fig. 4b, resembling the valve design from our previous work on magnetically controlled capsules[10]. Unlike single-layer magnetic soft valves[15], this design enables the valves to self-close in the absence of an external magnetic field, driven by the attractive gradient magnetic force between the valve frame and the leaf. This mechanism helps prevent potential liquid backflow after the magnetic field is removed. When an upward or downward magnetic field is applied, the valves open to the left or right, respectively. This functionality aligns well with the shape changes of the soft actuator, enabling directional pumping.

Given the bistable nature of the magnetoactive soft actuator, to reduce energy consumption during the application of alternating magnetic fields, we proposed an actuation strategy, as illustrated in Fig. 4c (i). This strategy involves applying a relatively high magnetic field during the shape transition phases of each pumping cycle, while reducing the magnetic field to a lower value during the remaining phases, allowing the actuator to maintain its shape due to its bistable properties. This represents a clear departure from traditional actuation waveforms applied to the actuator's current (i.e., the magnetic field waveform)[15,46]. In this context, we introduced a magnetic field-related parameter, defined as duty ratio $\eta = B_{keep} / B_{AMP}$, where $B_{keep}$ is the magnetic field maintenance amplitude and $B_{AMP}$ is the maximum magnitude of magnetic field, to modulate the current waveform. The effects of this modulation on the actuator's deformation are compared in Supplementary Information, Fig. S6, demonstrating that the waveform after energy-saving modulation has little impact on the actuator's deformation. It is noted that to ensure the effective operation of the magnetic control valve (detailed in Supplementary Information, Fig. S7), the duty ratio is typically set above zero during the pumping cycle, as depicted in Fig. 4c(i).

The effectiveness of this waveform design is demonstrated by the average gas pumping flow shown in Fig. 4d. For $\eta \geq 0.2$, the average pumping flow of the soft pump exceeds 90% of that achieved with a standard sinusoidal waveform at the same magnetic field amplitude. Notably, the average equivalent current (magnetic field) amplitude of our actuation waveform is only 73.49% of that of the sinusoidal waveform for $\eta = 0.2$, leading to a reduction in energy consumption (Supplementary Information, Note S3). Additionally, the magnetic field frequency can be easily adjusted under electromagnetic driving mode, which has a substantial impact on the pumping flow. Experimental results depicted in Fig. 4c (ii) reveal that increasing the magnetic field frequency enhances the number of compression cycles per unit time, thereby increasing the average gas pumping flow to a maximum of 228.1 mL/min at 3 Hz and 100 mT. Meanwhile, as both the magnetic field frequency and intensity rise during gas pumping, the output pressure of the magnetoactive soft pump gradually increases, peaking

at 2.525 kPa in the case of 3 Hz and 200 mT (Fig. 4d). By employing these energy-saving driving strategies, we sequentially inflated four 16-inch HUST letter balloons (considerably larger than the actuator size) using the soft pump (Supplementary Movie 6), demonstrating its potential in gas transport applications.

Additionally, we demonstrated the stable performance of the magnetoactive soft pump in liquid transport. The experimental setup, illustrated in Fig. 4f (i), shows the soft pump effectively delivering a Ponceau solution with consistent stability (Fig. 4f (ii) and Supplementary Movie 7). By adjusting the frequency and intensity of the applied magnetic field, we achieved a maximum average flow rate of 33.96 mL/min for pumping pure water in the experiments (Fig. 4g). Compared to similar existing magnetoactive soft diaphragm pumps in the Supplementary Information, Table S2, and other miniaturized soft pumps described in the work reported by Zhou et al.[47], our developed soft pump (>30 mL/min) achieves relatively high liquid output flow rates. Notably, our approach also achieves a substantial reduction in the required driving magnetic field, resulting in a higher normalized shape change ratio during pumping and an improved flow rate per unit magnetic field compared to existing works (Supplementary Information, Table S2). It should be noted that our magnetoactive soft pump can achieve effective pumping even at higher frequencies than those shown in Fig. 4g (e.g., at 10 Hz and 20 mT sinusoidal field, with a liquid pumping rate of 10.5 mL/min, as shown in Supplementary Information, Figure S8, and Supplementary Movie 8). However, due to limitations of the current power supply, the achievable magnetic field amplitude is restricted, and thus, a more in-depth study of pumping performance at these higher frequencies is not conducted in this work. Furthermore, the magnetoactive soft pump can maintain stable output performance across a range of liquid viscosities, from 0 to 40.8 cP, as measured using a viscometer, demonstrating its suitability for various liquid pumping applications (Fig. 4h and Supplementary Movie 9). Furthermore, we integrated a liquid flow sensor to achieve a closed-loop constant flow rate output for liquid media, as illustrated by the control logic in Fig. 4i. Using pure water as the test medium, we set a target flow rate curve (20 mL/min at point 2, 30 mL/min at point 4, and 0 mL/min at point 6), as shown in Fig. 4j. A microcontroller unit utilized feedback signals from the flow sensor to adjust the input current amplitude to the coil via a proportional integral (PI) controller in a closed-loop control system (Fig. 4k). The actual output flow rate closely followed the target curve (Fig. 4j), demonstrating precise closed-loop pumping control (Supplementary Movie 10). These results confirm the magnetoactive soft pump's capability for maintaining a consistent flow rate and reliable closed-loop flow control in practical applications.

## Reprogrammable magnetoactive metamaterials

Metamaterials are man-made structures with distinct physical properties through the meticulous design of microstructural arrangements[48]. The diversity and customizability of these properties make metamaterials applicable across various fields. In recent years, as technological demands have continued to rise, the reprogrammability of metamaterials has become an important research direction[30,49,50]. In response, we proposed a magneto-mechanical metamaterial design featuring shape reprogrammability at the unit-cell level, achieved through an array of our magnetoactive bistable soft actuators. Such metamaterial typically are well suited for shape reconfiguration and property tuning by applying an external magnetic field, enabling rapid, reversible, and untethered actuation[51,52]. Our design can be conceptualized as an array of physical binary elements, where each element can independently and reversibly switch between the two equilibrium states of a bistable shell. Compared to the continuous-type shape display[53,54], the binary-type presented in this work, while less effective at achieving smooth deformation control, is characterized by a straightforward switching mechanism, stable performance, and

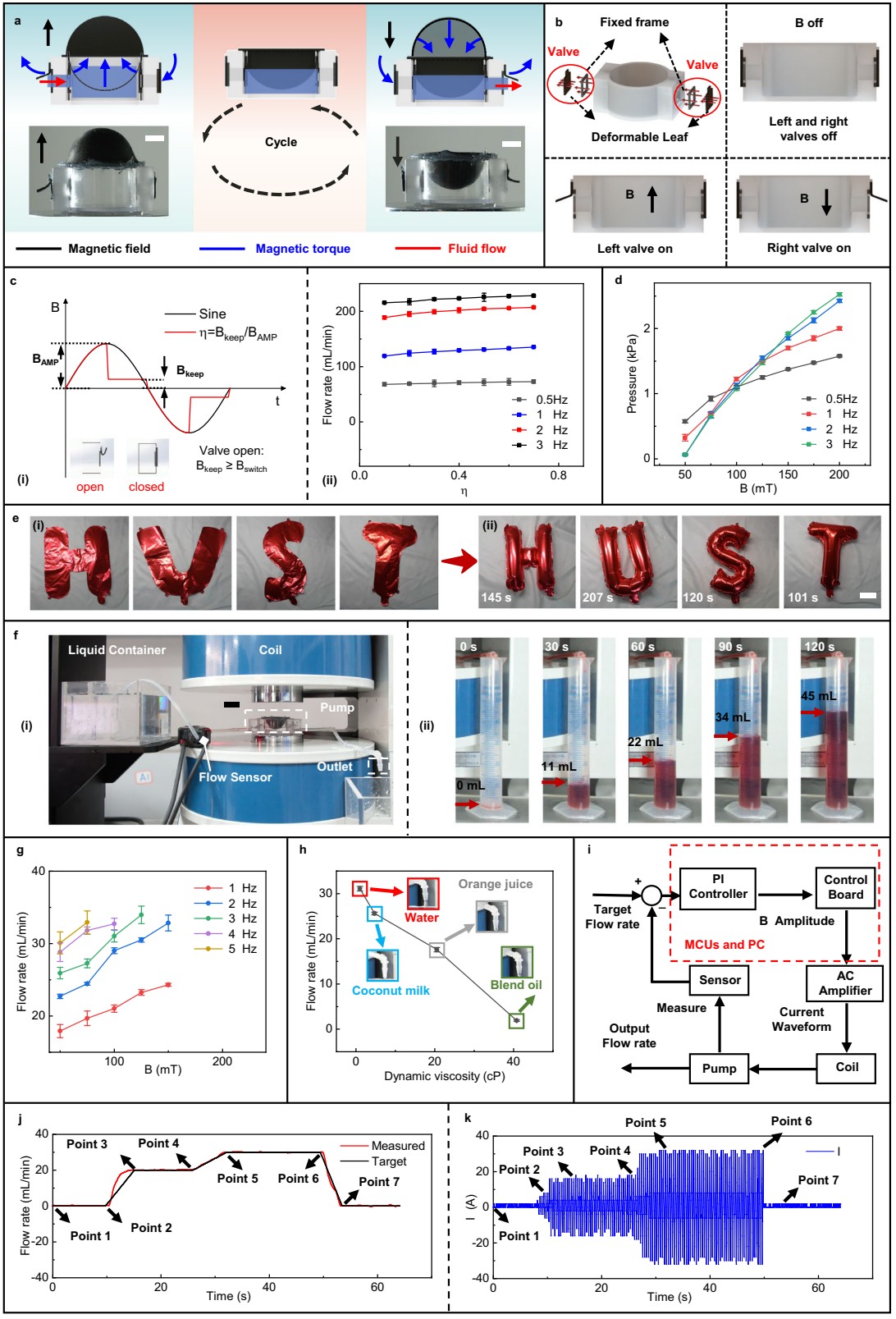

reduced control complexity. Furthermore, although our metamaterial operates in a manner similar to the approach by Chen et al.[30], there are clear differences in the magnetic structure design and actuation methods. Our work integrates magnetic components with mechanical motion components through magnetic soft materials, offering higher integration and simpler structures, which could provide a route todynamic control over the mechanical properties of metamaterials.

To validate the functionality of our shape-programmable metamaterial design, we constructed a prototype using actuator units with centimeter-scale diameters. We first tested actuator samples, which also serve as the metamaterial's magnetic units, with thicknesses of 0.5 mm and radii of 5, 7, and 9 mm to determine the switching magnetic field and response time in different cases (Fig. 5a). The data indicate that larger units require lower switching magnetic fields, and

**Fig. 4 | Magnetoactive soft pump utilizing bistable soft actuators. a** Pumping principle and experimental image of the magnetoactive soft pump. Scale bar: 10 mm. **b** Structure and working principle of the magnetic valve, consisting of a valve frame and valve leaf. Each pump incorporates two sets of these magnetic valves. **c** Average gas pumping flow rate of the magnetic soft pump under an energy-efficient drive waveform strategy. (i) Overview of the energy-saving drive strategy; (ii) Average gas pumping flow rates at different magnetic field frequencies and duty cycles ($n = 3$, data are presented as mean values ±SD). **d** Maximum gas pumping pressure achieved by the pump at various magnetic field frequencies and amplitudes ($n = 3$, data are presented as mean values ±SD). **e** Application of the magnetoactive soft pump for inflating HUST balloons. (i) Initial state of the balloon; (ii) Balloon state after inflation. Scale bar: 10 cm. **f** Application of the magnetoactive soft pump for liquid pumping. Scale bar: 20 mm. (i) Physical representation of the liquid pumping setup; (ii) Measured results of ponceau solution pumping. **g** Average liquid pumping flow rate at varying frequencies and amplitudes of the magnetic field ($n = 3$, data are presented as mean values ±SD). **h** Liquid pumping flow rate for different liquid viscosities ($n = 3$, data are presented as mean values ±SD). **i** Application of the magnetoactive soft pump for closed-loop precise flow rate control. **j** Time curve of the output flow rate versus the target flow rate. **k** Current amplitude curve during closed-loop control. Source data are provided as a Source Data file.

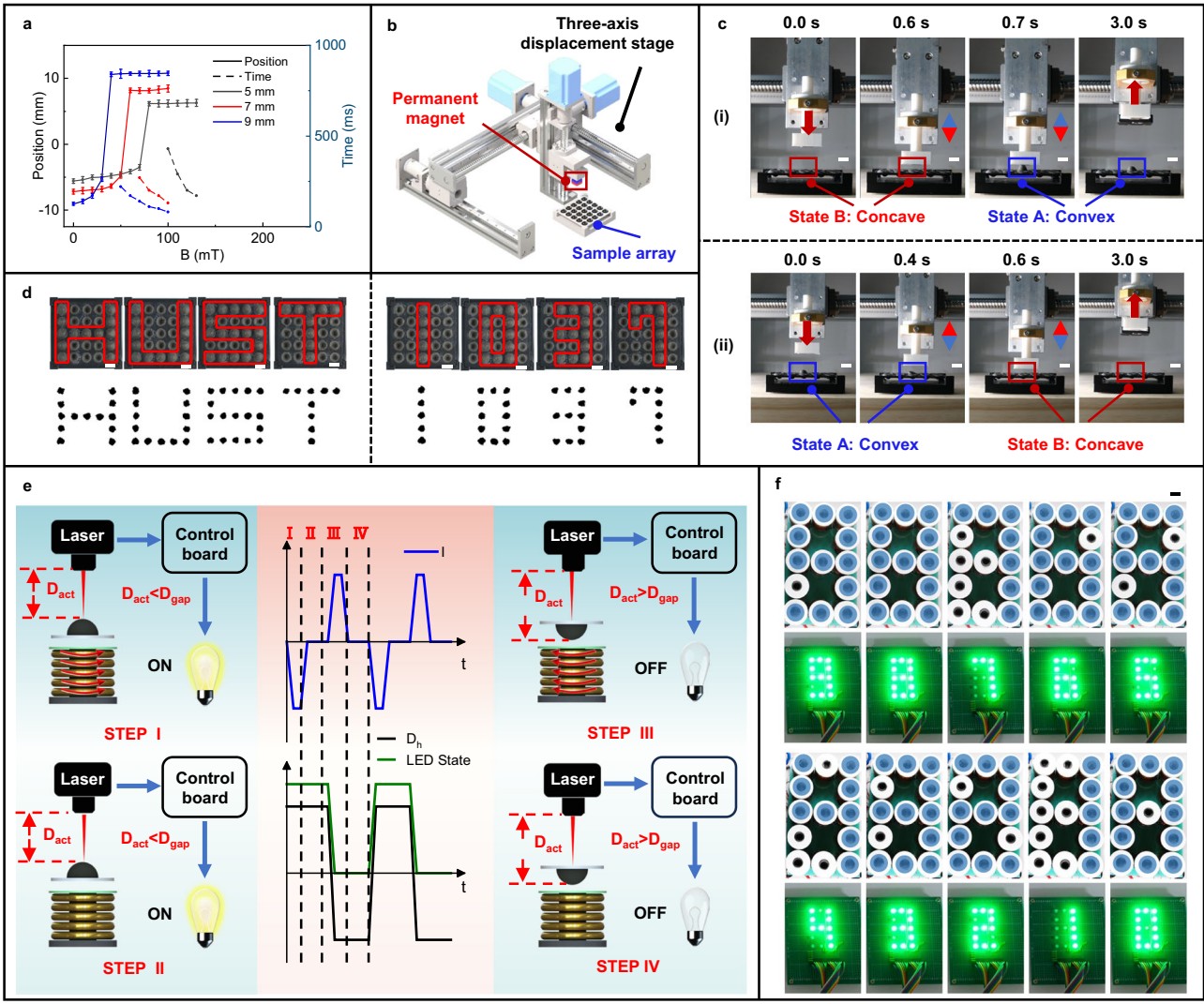

**Fig. 5 | Reprogrammable magnetoactive metamaterials utilizing bistable soft actuators. a** State switching processes and switching times of actuator samples of different sizes under varying magnetic fields. Sample thickness is 0.5 mm ($n = 3$, data are presented as mean values ± SD). Source data are provided as a Source Data file. **b** Schematic diagram of the permanent magnet actuation system for the magnetoactive metamaterials. **c** Shape and information programming of the metamaterials using the permanent magnet actuation system: (i) writing process, (ii) erasing process. Scale bar: 10 mm. **d** Programming display examples using the permanent magnet actuation system: (i) display of the "HUST" letter array, (ii) display of the "1037" number array. The actuator units have an $R$ of 5 mm and $h$ of 0.5 mm. Scale bar: 10 mm. **e** Diagram illustrating the programming and monitoring principles of a metamaterial unit using an electromagnetic actuation system. $D_h$ represents the height of the central point of the shell relative to the fixed base, where a positive value indicates an upward (convex) deformation and a negative value indicates a downward (concave) deformation. **f** Countdown display example using the electromagnetic actuation system, including images of the metamaterials and results from monitored emitting LEDs. The metamaterial units have an $R$ of 9 mm and $h$ of 0.5 mm. Scale bar: 9 mm.

within the same magnetic field, units with larger radii exhibit faster response times. As the magnetic field strength increases, the response time for units of identical size decreases noticeably. Notably, the response time for state change can be reduced to below 100 milliseconds. For instance, the response time of the 9 mm radius units decreased to 80 ms under a 100 mT magnetic field, highlighting the system's potential for high-frequency operations. Building on these results, we developed two metamaterial configurations with magnetic units of varying sizes for functional demonstration. The necessary magnetic fields for actuating the metamaterials were supplied by a

permanent magnet mounted on a robotic arm and an electromagnetic coil array, respectively.

As depicted in Fig. 5b, the permanent magnet actuation system consists of a square permanent magnet (N35, $10 \times 10 \times 20$ mm) and a three-axis mechanical displacement stage. By adjusting the position and magnetization direction of the magnet, the morphology of each actuator unit (concave or convex) can be precisely controlled, allowing for reversible switching between two stable states. Here, we defined the transition from a concave to convex state as the "writing" process, and the reverse transition as "erasure." This system allows for the reconfigurable programming of surface morphology in a metamaterial, featuring a 5 mm unit radius within a $5 \times 5$ array structure. The "writing" and "erasure" processes are fully controllable and repeatable, enabling the generation of diverse and customizable surface patterns. This capability was experimentally validated in Fig. 5c, where switching times of 0.1 s for writing and 0.3 s for erasure are demonstrated (Supplementary Movie 11), highlighting its rapid deformation response. Additionally, we successfully wrote and displayed the letter combination 'HUST' and the numerical information '1037' on the metamaterial. These features are clearly visible under high-visibility conditions, while in low-visibility environments, they can be identified by touch or captured using a magnetic imaging card, enabling easy retrieval of the displayed information from the card image (Supplementary Information, Figure S9 and Supplementary Movie 12).

Compared to permanent magnet-actuation methods, electromagnetic actuation systems offer improved dynamic control capabilities for the developed magnetoactive metamaterials, as they do not require mechanical movement or reversal of magnetic poles and offer faster adjustment rates. For demonstration purposes, we have self-developed an electromagnetic actuation system consisting of a coil array with 13 coils, along with power and control components (see Supplementary Information, Figure S10). This actuation system enables precise control of the magnetic field characteristics by adjusting the magnitude and direction of the current in each coil, facilitating the switching between convex and concave states of the actuator unit around the coil (STEPS I and III in Fig. 5e). When the coil current is zero, the magnetic unit maintains a stable convex or concave state due to its bistable nature (STEPS II and IV). Additionally, a laser monitoring system captures real-time information about the surface profile of the metamaterial and converts it into a visual output via an light emitting diode (LED) array that mirrors the same configuration. Specifically, the laser module measures the distance at the center of the unit's spherical surface to establish a switching distance threshold, $D_{gap}$, between the convex and concave states. When the unit is in a concave state (i.e., measured distance $D_{act} >$ threshold $D_{gap}$), the LED light turns off; conversely, when the unit is in a convex state (i.e., measured distance $D_{act} <$ threshold $D_{gap}$), the LED light illuminates. Using the above actuation method and control logic, we successfully executed a 9 s countdown from 9 to 0 (Supplementary Movie 13), employing a metamaterial with a digital display design, consisting of 13 bistable actuator units with a radius of 9 mm, and a corresponding 13-coil array, further demonstrating the applicability of our magnetoactive bistable soft actuators in the construction of multimodal metamaterials and their morphological control.

## Magnetoactive biomimetic soft grippers

The magnetoactive hemispherical soft actuator we proposed can switch between concave and convex bistable states, mimicking a motion similar to inhalation or exhalation, which enables potential applications in soft grippers, enabling the interaction of grasping and releasing external objects. Based on this, we constructed a soft gripper system with the soft actuator as the main structural component. Inspired by sea anemone structures, we added soft tentacle-like features to the smooth magnetic soft material surface to enhance the interaction between the gripper and the object, as shown in Fig. 6a.

The detailed structural parameters of the gripper are provided in Supplementary Information, Fig. S11, where we selected a tentacle length of 5 mm as an optimal design parameter for the gripper. This choice is supported by the discussion of the structural parameters' impact on gripping performance in Supplementary Information, Note S4 and Fig. S12, with a particular emphasis on how the tentacle's length and stiffness affect the gripper's performance.

Following axial magnetization, changing the direction of the applied axial magnetic field enables magnetic switching between the release state (State A) and the grasping state (State B). The soft gripper exhibits three gripping modes for objects of different sizes: (1) complete envelopment for small targets (Mode A); (2) multi-point grasping for medium-sized targets using magnetically controlled soft columns (Mode B), both requiring only a pulsed magnetic field; and (3) for large, heavy objects, a pulsed magnetic field induces concavity, followed by a sustained magnetic field that stretches the shell into an ellipsoidal shape, enhancing the grasping force (Mode C).

To enable flexible 3D object manipulation, a three-axis displacement platform was employed to control the spatial position of the gripper. Additionally, an electromagnetic coil was integrated above the gripper to provide the necessary actuation, as shown in Fig. 6b. During the grasping process, the gripper first approached the target object and then an upward magnetic field was applied to induce inward curvature of the gripper, allowing it to grasp the object. The bistable nature of the gripper allows it to maintain a stable grip in the absence of an external magnetic field after grasping. The object was subsequently released by applying a downward magnetic field (Fig. 6c, d, and Supplementary Movie 14). It is worth noting that the time required for the gripper to switch from a concave to a convex shape (defined as the convex delay) is longer than the time needed for the reverse process (concave delay), as illustrated in Fig. 6d. This phenomenon is primarily due to the gradient magnetic field generated by the coil above the gripper, which creates an attractive force between the gripper and the coil, leading to the observed delay.

Further experiments evaluated the gripper's performance with cylinders of various diameters and hexahedrons of different masses. The gripper efficiently handled objects with diameters ranging from 7 to 17 mm, achieving a success rate of over 60%, and 100% success for objects between 9 and 15 mm (Fig. 6e). The gripper's grasping ability is closely related to the mass of the object. Taking an object with a diameter of 13 mm as an example, experimental results show that without applying additional current, that is, relying solely on the bistable characteristic, it becomes difficult to complete the grasping action when the object's weight exceeds 1.5 g, as shown in Fig. 6f. This issue can be addressed by applying current to the coil above the gripper to generate an auxiliary magnetic field, thereby increasing the gripper's stiffness. Experimental results demonstrate that with a 3 A current, the effective grasping mass can be increased to 3.5 g, increasing its capability (Fig. 6f). This also demonstrates the adjustability of the magnetic bistable actuator. Furthermore, to investigate the effect of magnetic field strength on the gripper's stiffness, we conducted experiments using the setup in Fig. 6g. The gripper grasped a small sphere ($R = 7$ mm) attached to a force gauge, with the vertical magnetic field strength controlled by varying the coil current. A displacement stage pulled the sphere downward along the Z-axis. Results in Fig. 6h show that the gripping force increased to 29 mN as the coil current rose from 0 to 4 A. At lower currents ($I \leq 2$ A), the maximum gripping force occurred at $d = 1$ mm, likely due to resistance from the tentacle motion. At higher currents ($I \geq 3$ A), the maximum gripping resistance (at $d = 0$ mm) increased with the coil current. Moreover, the compression resistance in Fig. 6i also increased with coil current due to the upward magnetic torque on the hemispherical shell, which prevented downward deformation after gripping and improved object stability. These findings confirm that adjustments in the magnetic field

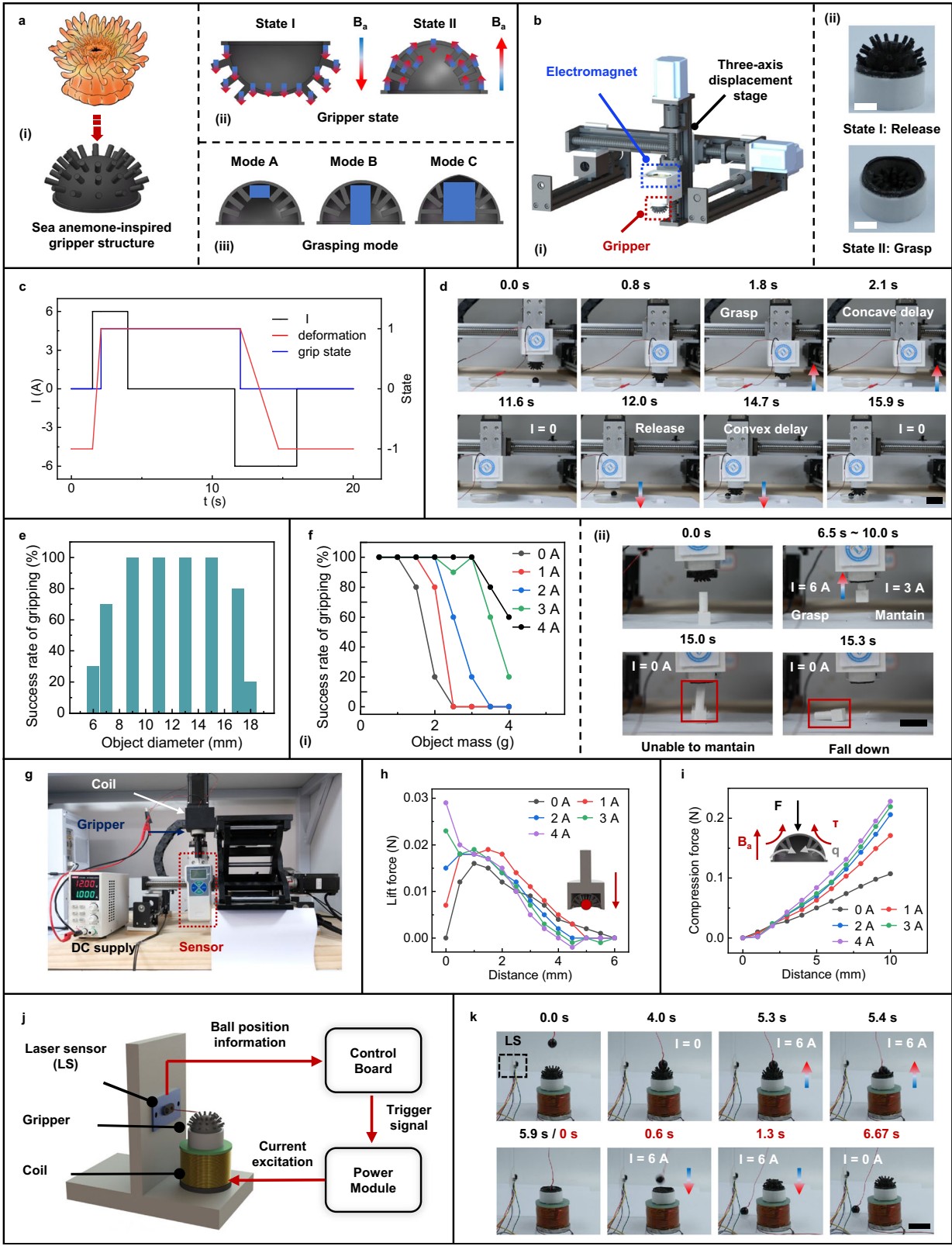

effectively control the gripper's stiffness, optimizing its maximum gripping performance. Finally, we integrated a laser sensor and controller into the soft gripper, enabling environmental self-recognition and autonomous grasping (Fig. 6j and Supplementary Information, Fig. S13). The system measured the distance to an object to determine if it was within range, then initiated grasping or releasing actions based on the received signals, effectively demonstrating automatic recognition as well as self-grasping and releasing capabilities (Fig. 6k and Supplementary Movie 15).

## Discussion

In this work, we present a magnetoactive actuator that integrates magnetic actuation, soft materials, and bistable mechanics within a 3D hemispherical shell embedded with NdFeB microparticles. This

**Fig. 6 | Magnetoactive biomimetic robotic gripper utilizing bistable soft actuators. a** Design of the robotic gripper inspired by sea anemone structure: (i) structural design, (ii) stable states, and (iii) grasping modes. **b** Schematic diagram of the 3D biomimetic sea anemone gripper system (i) and experimental images of the gripper (ii), with a diameter of 30 mm. Scale bar: 10 mm. **c** Control waveform of the gripper system during the grasping process. **d** Experimental images showing the gripper grasping and releasing a small ball. Scale bar:30 mm. **e** Grasping success rate of the gripper for cylindrical objects with varying diameters. **f** Grasping capability for cylindrical objects of different masses: (i) success rate for cylindrical objects under varying magnetic fields, (ii) experimental images of the gripper

grasping a 3 g object. Successful grasping occurs at 6 and 3 A currents but fails to hold at 0 A. Scale bar: 30 mm. **g** Experimental setup for measuring the gripper's grasping force under an external magnetic field. **h** Relationship between grasping force and external magnetic field strength. **i** Relationship between compression resistance and external magnetic field strength. Source data are provided as a Source Data file. **j** Schematic diagram of the gripper system with environmental recognition and autonomous grasping capabilities. **k** Experimental images showing the gripper's self-recognition grasping and active release of a small ball. The black ball has a mass of 1.5 g and a diameter of 14 mm. Scale bar: 30 mm.

design overcomes the strong correlation between deformation amplitude and the applied magnetic field strength, a limitation commonly observed in conventional magnetoactive soft actuators. The actuator enables efficient switching between bistable states under low-strength pulsed magnetic torque disturbances, while remaining stable in the absence of an external magnetic field. This advancement addresses the persistent challenge of relying on continuous and relatively high magnetic fields (>100 mT) to sustain large-scale shape transformations, particularly those with shape change ratios exceeding 0.5 and occurring under constrained boundary conditions. Compared to existing magnetoactive soft actuators (Supplementary Information, Tables S1 and S3, and Supplementary Movie 16), our design achieves large-scale shape transformations with a shape change ratio exceeding 0.8 under magnetic fields below 20 mT, even as low as 10 mT (Supplementary Information, Table S1, and Fig. S14), thereby exceeding previously reported performance benchmarks in this field. Furthermore, we demonstrate the versatile applications of our magnetoactive bistable actuator through three representative applications: a high-efficiency soft pump, a reprogrammable metamaterial, and a variable-stiffness soft gripper. These applications not only validate the actuator's capability for large, reversible deformations under low magnetic fields, but also highlight its potential in enabling closed-loop fluid control, logic-embedded 3D architectures, and adaptive, stiffness-tunable grasping. Overall, this work provides insights into magnetoactive soft actuator technology and could inspire further exploration of bistable and multistable structures in the field of soft actuators.

Looking ahead, several key areas need further exploration. First, optimizing key parameters including shell geometry, material stiffness, and magnetic design (such as spatial magnetization profiles and field alignment) is essential for further reducing the magnetic field required for actuation. However, due to the inherent complexities of magneto-mechanical coupling, nonlinear large deformation, and sensitivity to boundary conditions, it remains challenging to accurately predict the critical switching torque or the required magnetic field. Therefore, developing theoretical models and accurate simulations that capture these effects will play an important role in guiding future actuator design. Second, miniaturizing bistable soft actuators to potentially sub-millimeter scales is crucial for their integration into compact systems where high spatial resolution is needed. Achieving this will require advances in fabrication techniques and innovative strategies to further enhance magnetic field responsiveness. Third, although this work primarily focuses on shape control, the full potential of the proposed metamaterial configuration remains to be explored. Future studies should investigate its mechanical, optical, and acoustic properties to unlock applications in flexible electronics, advanced optics, and multifunctional sensing technologies. Lastly, the energy-efficient switching behavior enabled by magnetic bistability presents promising opportunities in energy amplification systems. Exploring the role of bistability in energy storage, mechanical amplification, and dynamic reconfigurable structures could lead to further progress in high-efficiency actuators, energy-harvesting devices, and smart materials.

## Methods

### Preparation and characterization of magnetoactive bistable soft actuators

The preparation process of magnetoactive bistable soft actuators mainly consists of material mixing, molding, demolding, and magnetization (Fig. 2a). NdFeB particles (5 μm, MQP-15-7, Tianjin Magnequench Co., Ltd., China) and silicone elastomer (Ecoflex 00-20, Smooth-On, Inc.) were mixed in a 1:1 mass ratio. The mixture was stirred at 2000 r.p.m. for 90 s, then defoamed at 2200 r.p.m. for 60 s in a planetary mixer. The resulting magnetic slurry was poured into a concave Polylactic Acid (PLA) mold printed by a 3D printer (AD5M, Zhejiang Flash Casting Group Co., Ltd., China). A PLA convex mold was then pressed onto the concave mold, allowing excess slurry to flow out through the channels. A weight was placed on the convex mold to ensure proper alignment. After 3 hours at room temperature, the magnetic composite was fully cured. The actuator sample was then demolded and magnetized using a pulsed high magnetic field.

To evaluate the bistable properties of the soft actuators with different geometries, the sample edges were adhered to a circular mold using food-grade silicone adhesive (Fig. 2c(ii)). A 5 mm diameter PLA rod was used to press the samples from a convex to a concave state. After holding the pressure for a set time, the rod was gently removed to assess whether the samples maintained the concave shape without external force. If stable, the actuator was deemed bistable. Additionally, the effect of different substrate materials on the compressive resistance of soft actuators was also investigated (Fig. 2c). A push-pull force gauge (SF-5, Hangzhou Airep Electronic Technology Co., Ltd., China) mounted on a three-axis displacement platform (range: 200 mm, FGWISDOM, China) was used to compress soft actuator samples ($R = 10$ mm, $h = 1$ mm) with a 1:1 mass ratio of magnetic particles to substrates, including silicone elastomer (Ecoflex 00-20, Smooth-On, Inc.) and PDMS (SYLGARD 184, Dow Corning Co., USA). Compression resistance per unit distance was measured throughout the process. Due to the gauge's range of 0.5 N to 5 N, a printed PLA mold was attached to the gauge, applying an initial compression force of 0.7 N. This allowed for successful measurement of the compressive force curve with a 0.7 N offset, with each curve measured three times.

Figure 2d, e show experiments conducted to determine the magnetic field required for state switching in soft actuators with varying magnetic powder content and particle sizes. The uniform magnetic field was generated by an electromagnet (WD-80V-68AC, Changchun Yingpu Magnetelectric Technology Co., Ltd., China), and the field intensity was measured with a Gauss meter (TD8620-1, Changsha Tianheng Measurement and Control Technology Co., Ltd., China). The deformation was recorded by a camera (X-H2, Fujifilm, Japan). For repeatability, three samples of the same radius and thickness were prepared (Fig. 2e).

### Magnetic response characteristics of magnetoactive bistable soft actuators

In the magnetoactive bistable soft actuators' magnetic response performance test shown in Fig. 3, the DC current in the electromagnetic actuation system was supplied via a DC power source

(SP40VDC2000W, APM Technologies Co., Ltd., China), while the AC current was controlled by an AC amplifier (HEA-500G, Changchun Impex Magneto Technology Co., Ltd., China). The AC current was generated by a controller (STM32F407GTZ, STMicroelectronics, Italy) and a Digital-to-Analog Converter (DAC) module (DAC8563, Texas Instruments, USA), which produced an arbitrary waveform that was then amplified to create the corresponding magnetic field waveform. A laser displacement sensor (ZW-LV100R-NP, Shenzhen Jingjiake Intelligent Sensing Co., Ltd., China) measured the variation in the Z-direction coordinate at the center of the spherical shell over 100 consecutive cycles. As shown in Fig. 3e and f, the magnetic actuation process under DC and sine wave AC currents (frequency: 1–6 Hz, increment: 1 Hz) was recorded using high-speed photography (300 frames/s, X-H2, Fujifilm, Japan).

### Preparation and performance evaluation of magnetoactive soft pumps

The magnetoactive soft pump consists of two magnetic valves, a soft actuator unit, and a transparent pump container. Each valve includes a valve frame and a valve leaf, both made from magnetic composite material. The composite slurry, prepared in the same manner as for the soft actuator, has a 1:1 mass ratio of magnetic particles to silicone elastomer. The slurry was poured into printed PLA molds for the valve frame and leaf, cured at room temperature for 3 h, then demolded and magnetized. The soft actuator unit and valves were then affixed to a transparent resin housing (Shenzhen Chennuo 3D Technology Co., Ltd., China) using food-grade silicone adhesive to form the pump (Fig. 4a).

To assess the effect of the magnetic field waveform on pumping performance, the duty cycle and frequency of the sinusoidal magnetic field were varied (Fig. 4c). Gas pumping pressure was measured with a pressure module (XGZP6847A 0-5 kPa, Wuhu Core Silicon Smart Electronic Technology Co., Ltd., China). Instantaneous gas and liquid flow rates were recorded using a gas flow sensor (FS4001-500-CV-A, Silicon Microelectromechanical Systems Co. Ltd., USA) and a liquid flow sensor (FD-XS1, KEYENCE, Japan), respectively. Waveforms were captured with an oscilloscope, and the average flow rate over five cycles was calculated, with error bars determined from multiple cycles. In the pumping performance evaluation of different types of liquids (Fig. 4h), the viscosity of the liquids was measured using a viscometer (AS-NDJ-5S, Shanghai EXCEL Technology Co., Ltd., China).

In the closed-loop flow control experiment (Fig. 4j), instantaneous flow was measured with a liquid flow sensor (FD-XS1, KEYENCE, Japan). A resistance sampling module ($R_L = 150\Omega$) converted the analog current signal (0-20 mA, flow range 0–100 mL/min) into an analog voltage signal (0–3 V) for sampling by the Analog-to-Digital Converter (ADC) module on the control board (STM32F407GTZ chip, STMicroelectronics, Italy). PI parameters were adjusted as per Fig. 4i to complete the closed-loop control. The target flow rate was adjusted in real time, with flow and coil current waveforms captured by an oscilloscope.

### Design and display testing of reprogrammable magnetoactive metamaterials

The uniform magnetic field for data analysis in Fig. 5a was generated by an electromagnet (WD-80V-68AC, Changchun Yingpu Magnetelectric Technology Development Co., Ltd., China). The magnetic induction intensity was adjusted by altering the output current of the DC power supply, and the deformation process was photographed by using the high-speed mode (300 frames/s) of the camera (X-H2, Fujifilm, Japan). The position of each soft actuator sample and the response time of the steady-state switching process were obtained from the video. The messages "HUST" and "1037" in Fig. 5d were programmed and realized using a three-axis mechanical displacement stage (range: 200 mm, FGWISDOM, China) equipped with a permanent magnet. The LED display in Fig. 5 was achieved through the electromagnetic actuation system shown in Supplementary Information Figure S10. This system consists of 13 solenoid coils with iron cores (outer diameter 50 mm, height 40 mm, with a magnetic field strength exceeding 50 mT at 15 mm from the magnet surface during operation), 13 sets of soft actuator units (radius: 9 mm, thickness: 0.5 mm), and 13 light-emitting diodes, arranged in the form of digital tubes. A counting program with a 1 s interval was run on the host computer, and the control board adjusted the power supply to regulate the current for each coil, enabling the 9-second countdown from 9 to 0.

### Grasping ability test of magnetoactive soft grippers

The test objects in Fig. 6e were cylindrical parts of varying sizes (thickness: 5 mm, diameters: 6–18 mm), printed using a 3D printer. The mass of these objects was kept within ±0.1 g of 1 g by adjusting the printing fill parameters to ensure consistent weight across all objects. The current waveform of the electromagnet during the grasping process is shown in the 0–7.5 s portion of Fig. 6c, with the holding current set to 0 A. The experiment was repeated 10 times, and the success rate for grasping cylindrical targets of the specified diameters was recorded.

In the grasping performance test shown in Fig. 6f, the object mass was adjusted as follows: the fill rate was modified to achieve a mass of 0.5 g for cylindrical parts with a thickness of 5 mm and a diameter of 13 mm, while the thickness was varied to regulate the mass of other cylindrical parts (ranging from 0.5 g to 4 g in 0.5 g increments). The gripper attempted to grasp the target objects with holding currents set at 0, 1, 2, and 3 A. The experiment was repeated 10 times, and the success rate for grasping cylindrical targets under specified mass and holding current conditions was recorded.

In the self-recognition grasping experiment shown in Fig. 6k, a laser sensor (MS53L0M, Guangzhou Xingyi Electronic Technology Co., Ltd., China) measured the horizontal distance between the falling object and the sensor. When the distance was within the given range, the control board sent a signal to the power supply module (self-designed, Supplementary Information, Figure S13). The gripper then grasped the object upon receiving current. Pressing the release button sent a release signal to the power module, which applied reverse current to release the object, resetting the gripper for the next object.

## Data availability

The authors declare that the main data supporting the findings of this study are available within the article, Supplementary Information files and Source data. Extra data are available from the corresponding author upon request. Source data are provided with this paper.

## Code availability

The codes for operating the microcontroller board are available from the corresponding author upon reasonable request.

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

## Acknowledgements

We gratefully acknowledge the financial support of the National Natural Science Foundation of China (Grant No.52422701 (Q.C.)). We would like to thank Yansong Tao at the Wuhan National High Magnetic Field Center for assistance with characterization and for proofreading the manuscript.

## Author contributions

Conceptualization: Q.C. and L.L.; Methodology: Q.C., H.W., Y.S., Z.S., and L.L.; Investigation: H.W., Z.S., Y.S., L.X., X.Z., C.M., and F.X.; Visualization: H.W. and Z.S.; Funding acquisition: Q.C.; Project administration: Q.C. and L.L.; Supervision: Q.C. and L.L.; Writing—original draft: H.W. and Z.S.; Writing—review and editing: Q.C., H.W., Z.S., and X.L. All authors contributed to the discussion.

## Competing interests

The authors declare no competing interests.
