## [Transparent Peer Review file · Nature Communications]

Magnetoactive Bistable Soft Actuators for Programmable Large Shape Transformations at Low Magnetic Fields

Corresponding Author: Professor Quanliang Cao

Version 0:

Reviewer comments:

Reviewer #1

(Remarks to the Author)

In this work, the authors experimentally studied the static and dynamic behaviors of hemispherical bistable magnetic actuators and used the novel actuators to design intriguing applications, such as a magnetic pump. The topic is engaging and informative, and the study has the potential to contribute significantly to this field. While the manuscript presents valuable insights, there are some concerns and areas for improvement that need to be addressed in a revision.

Main Issues and Concerns:

Text S1

Text S1 is quite challenging to follow and could benefit from significant refinement:

The magnetization of the hemispherical actuators in this study is aligned with the z-direction in the concave state, which differs from the actuators described in (Chen et al., 2024, Adv. Funct. Mater). Consequently, the principle used in the explanation is not directly applicable here. Instead, the actuators in the concave state under a reversed magnetic field resemble an unstable equilibrium, much like a compressed rod. When the magnetic field exceeds a critical value, the actuators rapidly transition from a zero-strain state to a bending state.

Additionally, not all geometric parameters are clearly labeled in the diagram, and there are inaccuracies in its description. For instance, while the actuators are axisymmetric with the Z-axis as the symmetric axis, line 32 describes them as having unit thickness along the X-axis, which seems inconsistent.

There are also several errors in the equations. For example, the dot product in Eq. 1 should be a cross product, and the simplification in Eq. 3 is incorrect. Furthermore, the dot product symbols are not displayed properly on the first page of Text S1.

Given these issues, I suggest reconsidering the necessity of Text S1 in this manuscript. If it is not critical to the central narrative, it might be more effective to remove it altogether.

Line 107

The statement about the gripper adjusting its stiffness to handle heavy objects is compelling but would be greatly strengthened by including quantitative analysis. I encourage the authors to provide such analysis in Section 2.5.

Section 2.1

The stability boundary model provided to determine bistability for samples with varying geometric parameters is well-executed, and the experimental results shown in Fig. 2(b) align nicely with predictions. To make this concept even more accessible, particularly for readers who may not specialize in mechanics, including a video showing the deformation process of the two types of samples would be highly beneficial.

Lines 139–147

The comparison of displacements between bistable and "conventional" actuators seems to overlook the design intent of conventional structures, which are meant to avoid large deflections. Instead, the focus should be on the advantages of bistable structures, such as their rapid configuration transitions and ability to maintain deformation under zero load. I recommend removing the current comparison and discussing these unique advantages in more depth.

Section 2.4

While the structures discussed in this section are fascinating, referring to them as metamaterials requires more detailed explanation. What specific types of metamaterials do they represent—mechanical, electrical, or photonic—and what unique properties qualify them as such? If these classifications and justifications cannot be clearly articulated, I encourage the authors to use a more appropriate term to describe these structures.

Section 2.5

The geometry of the bar in the gripper likely influences its grasping capability. I strongly encourage the authors to explore this aspect further through simulations or experiments, or at least discuss it more comprehensively in the manuscript.

Minor Comments and Suggestions:

Carefully proofread the manuscript to minimize typographical and grammatical errors.

Ensure consistent symbol formatting throughout the paper. For instance, while symbols in the supplementary material are italicized, those in line 118 are not. Adhering to conventional formatting rules (e.g., Times New Roman for Latin letters and italicized Greek letters for scalars) will enhance readability.

Introduce all abbreviations with their full names when they first appear in the text.

Consider using "wt%" instead of "mass fraction" for clarity, specially in figures

Adjust the placement of the scale bar in Fig. 2(d) for better visual alignment.

Reviewer #2

(Remarks to the Author)

The manuscript title "Magnetoactive Bistable Soft Actuators for Programmable Large Shape Transformations at Low Magnetic Fields by Quanliang Cao et al. introduces magnetoactive actuators made from silicon-based elastomers shaped in a dome structure. It demonstrates the potential for various applications, such as soft pumps, metamaterials, and soft grippers, due to its simple operating mechanism and fabrication process. Furthermore, the ability to achieve large deformations under low magnetic fields (and thus low energy consumption) appears to be a notable achievement in the field of magnetic actuators. However, since the publication of magnetic actuators by Metin Sitti's group[1], many studies on magnetic actuators have been conducted. This raises the question of whether simply creating a dome-shaped actuator truly adds novelty to the field. Additionally, creating soft pumps, metamaterials, and grippers is not fundamentally a new concept. There are also several concerns and questions regarding the manuscript. In the current stage, this manuscript is therefore not suitable for publication in Nature communications.

1. In the overall structure, the magnetoactive bistable soft actuators are made with a 50% magnetic particle composition of Ecoflex 0020 and a thickness of 0.5 mm. However, the state A without an applied magnetic field is not presented anywhere. Is this because the structure cannot maintain itself in that state? If so, PDMS might be a more favorable option.

2. The design and principle of the soft pump are strikingly similar to those in ref [2]. Additionally, the results in that reference are similar to, or superior to, those in this study in terms of the time required for liquid transport, the range of minimum magnetic fields, and the conversion frequency.

3. The papers cited as references for metamaterials focus on studies with a single unit cell [3]. However, the experiments in this paper are more akin to a 'meta-surface'. When considering the research scope of meta-surfaces, the application of magnetoactive bistable soft actuators is somewhat lacking in terms of lower resolution and binary stage changes (0 and 1) [4].

4. There are several issues with numbering and missing sections throughout the manuscript. For example, the order of Fig. 3b and 3c does not match the sequence in the manuscript, and there is no Results 2.3.

5. Furthermore, the manuscript repeatedly claims higher performance compared to previous work, but it is necessary to clearly state these comparisons with existing studies.

6. While the comparison in the Supplementary Information is more organized than expected, it lacks visibility, and some points are inaccurate (e.g., in the data from reference 9 in the supplementary information, when both membranes are applied, the average flow rate is higher than 7 mL/min).

[1] Hu, W., Lum, G., Mastrangeli, M. et al. Small-scale soft-bodied robot with multimodal locomotion. *Nature* 554, 81–85 (2018). <https://doi.org/10.1038/nature25443>

[2] Lin, D., Yang, F., Gong, D. & Li, R. Bio-inspired magnetic-driven folded diaphragm for 544 biomimetic robot. *Nat. Commun.* 14, 163 (2023). [3] Jiao, P., Mueller, J., Raney, J.R. et al. Mechanical metamaterials and beyond. *Nat Commun* 14, 6004 (2023). <https://doi.org/10.1038/s41467-023-41679-8>

[4] Johnson, B.K., Naris, M., Sundaram, V. et al. A multifunctional soft robotic shape display with high-speed actuation, sensing, and control. *Nat Commun* 14, 4516 (2023). <https://doi.org/10.1038/s41467-023-39842-2>

Reviewer #3

(Remarks to the Author)

Version 1:

Reviewer comments:

Reviewer #3

(Remarks to the Author)

Reviewer #4

(Remarks to the Author)

The authors demonstrated a magnetically driven soft bistable actuator that can maintain two stable states in the absence of a magnetic field and achieve shape changes in a low-intensity magnetic field. In addition, the authors demonstrated the applications of the developed actuator for pumping liquids or gases, programmable metamaterials for information encoding and soft grippers. However, the core concept of combining magnetic actuation with bistable structures has been extensively explored in prior literature. While the authors claim improvements in switching time and reduced magnetic field requirements (<100 mT), these metrics are not groundbreaking compared to recent literature. For example, similar performance has been achieved with other bistable designs (Pal & Sitti, PNAS, 120, e2212489120 (2023); Tang et. al, Sci. Robot. 9, eadm8484 (2024)). In addition, the theory and deformation of bistable spherical shells have been extensively studied (e.g. Sci. Robot. 5, eabb1967 (2020)). The demonstrated applications (soft pumps, metamaterials, grippers) are conceptually similar to existing systems. While the study is technically sound, it looks like lack of enough novelty and contribution to the advancement of this field, it does not meet Nature Communications's threshold for novelty and broad impact. The work appears incremental, with insufficient differentiation from prior art in bistable magnetic actuators. The research topic and approach are good, but scientific findings and technical improvement are required for the broad readership of the journal. Below are some comments on the manuscript for the author's reference:

- (1) Can the authors provide a more detailed comparison of the quantitative improvement in the driving magnetic field, and do more scientific parameterized comparison (Table S1)?
- (2) On page 3 line 72, "As a result, achieving large shape transformations under low magnetic fields remains largely unexplored, and the development of corresponding functional applications is still in its early stages." How do you define large deformation? What is the standard for low magnetic field strength? It would be more convincing if you could give specific parameters.
- (3) Are the currently designed driving magnetic fields (20 - 100 mT) close to the theoretical lower limit? Will the deformation speed and driving magnetic field strength be further reduced under smaller sizes and softer material systems? Is there a universal relationship between critical torque and shell stiffness/curvature?
- (4) Authors should carefully review their manuscripts before submission. On Page 10 line 296 - 307, the same content appears twice.

Additional mark:

The authors have fully addressed the concerns and comments raised by Reviewer 1 and have basically addressed those from Reviewer #2. However, some responses to Reviewer #2 would benefit from further revision:

- (1) Contribution to the field (response to Reviewer #2's overall comments): The authors' response regarding the study's contribution does not appear sufficiently convincing.
- (2) For the response to Reviewer #2's comment 5 (Table S2). While the authors revised Table S2 in response to this comment, I remain concerned that the "snap time" column does not provide a fair comparison. The response time of the different bistable actuator is closely related to factors such as material stiffness and actuator size.
- (3) Both Reviewers #1 and #2 mentioned issues with the rigor of the manuscript, especially typographical and grammatical errors (e.g., numbering, abbreviations, etc). Although the authors have made revisions, some lingering issues remain. For example, on Page 10 (line 296 - 307), the same content appears twice. The authors should carefully proofread the manuscript.

Version 2:

Reviewer comments:

Reviewer #4

(Remarks to the Author)

I appreciate the efforts from the authors to address my previous comments.

The quality of this work has been improved significantly after revisions, and I have no further questions. It is recommended for publication now.

Responses to Comments on “NCOMMS-24-64525”

Dear reviewers:

We express our sincere gratitude for your careful review of our manuscript. Those comments are all valuable and very helpful for revising and improving our paper, as well as the important guiding significance to our researches. According to all comments, the manuscript has been revised carefully. We believe that the modifications have contributed to a significant improvement of our manuscript, making it more suitable for publication in Nature Communications.

The main corrections within the paper have been highlighted using red text, and the responses to the comments are list within the subsequent text sections.

Sincerely Yours,

Prof. Quanliang Cao & Prof. Liang Li

Wuhan National High Magnetic Field Center, Huazhong University of Science and
Technology

Response to the Reviewer #1:

General comment: In this work, the authors experimentally studied the static and dynamic behaviors of hemispherical bistable magnetic actuators and used the novel actuators to design intriguing applications, such as a magnetic pump. The topic is engaging and informative, and the study has the potential to contribute significantly to this field. While the manuscript presents valuable insights, there are some concerns and areas for improvement that need to be addressed in a revision.

Response: Thank you for the positive feedback and suggestions. We will provide point-by-point response to the specific comments in the following part.

Comment 1: Text S1 is quite challenging to follow and could benefit from significant refinement: The magnetization of the hemispherical actuators in this study is aligned with the z-direction in the concave state, which differs from the actuators described in (Chen et al., 2024, Adv. Funct. Mater). Consequently, the principle used in the explanation is not directly applicable here. Instead, the actuators in the concave state under a reversed magnetic field resemble an unstable equilibrium, much like a compressed rod. When the magnetic field exceeds a critical value, the actuators rapidly transition from a zero-strain state to a bending state.

Additionally, not all geometric parameters are clearly labeled in the diagram, and there are inaccuracies in its description. For instance, while the actuators are axisymmetric with the Z-axis as the symmetric axis, line 32 describes them as having unit thickness along the X-axis, which seems inconsistent.

There are also several errors in the equations. For example, the dot product in Eq. 1 should be a cross product, and the simplification in Eq. 3 is incorrect. Furthermore, the dot product symbols are not displayed properly on the first page of Text S1.

Given these issues, I suggest reconsidering the necessity of Text S1 in this manuscript. If it is not critical to the central narrative, it might be more effective to remove it altogether.

Response 1: Thank you for your valuable feedback. The issues you raised are indeed valid, and we acknowledge the areas that need improvement. Regarding the unclear description and the errors in the equations in Text S1, we apologize for the oversight and appreciate your detailed comments. Following your suggestion, and considering that this section is not critical to the central narrative, we have removed it in the revised manuscript. Additionally, to explain the shape switching mechanism of our hemispherical actuator under the influence of a magnetic field, **we have added further descriptions in the main text of the revised manuscript.**

On page 4, we made revisions to the text:

“The shape transformation is driven by the torque induced by the misalignment between the magnetization of the hemispherical shell and the applied external magnetic field. Prior to the transformation, the applied magnetic field is aligned antiparallel (180°) to the magnetization direction, theoretically resulting in zero initial torque^[36]. However, even a slight perturbation—such as a small deviation in the local magnetization direction—can generate local magnetic torques^[37], amplifying non-zero torque regions and triggering a shape transition between the convex and concave states when the magnetic field exceeds a critical value.”

Comment 2: In Line 107, the statement about the gripper adjusting its stiffness to handle heavy objects is compelling but would be greatly strengthened by including quantitative analysis. I encourage the authors to provide such analysis in Section 2.5.

Response 2: Thank you for your insightful suggestion. In the original manuscript, we focused only on the effect of maintaining magnetic field strength (by varying the coil current) on the gripper's ability to handle heavy objects, without conducting a quantitative study on how magnetic field strength affects the gripper's stiffness. **In the revised manuscript, we have updated Section 2.4 (updated section number) to include a quantitative analysis of stiffness adjustment and its effect on gripping performance**, strengthening the discussion and providing a clearer understanding of the gripper's capabilities.

On page 14, we made revisions to the text (including the addition of Fig. 6g-i and related descriptions):

“To investigate the effect of magnetic field strength on the gripper's stiffness, we conducted experiments using the setup in Fig. 6(g). The gripper grasped a small sphere ($R=7$ mm) attached to a force gauge, with the vertical magnetic field strength controlled by varying the coil current. A displacement stage pulled the sphere downward along the Z-axis. Results in Fig. 6(h) show that the gripping force increased to 29 mN as the coil current rose from 0 to 4 A. At lower currents ($I \leq 2$ A), the maximum gripping force (17 mN) occurred at $d = 1$ mm, likely due to resistance from the tentacle motion. At higher currents ($I \geq 3$ A), the maximum gripping resistance (at $d = 0$ mm) increased with the current. Additionally, the compression resistance in Fig. 6(i) increased significantly with coil current due to the upward magnetic torque on the hemispherical shell, which prevented downward deformation after gripping and enhanced object stability. These findings confirm that adjustments in the magnetic field effectively control the gripper's stiffness, optimizing its maximum gripping performance.”

Fig. 6. Magnetoactive biomimetic robotic gripper utilizing bistable soft actuators. (g) Experimental setup for measuring the gripper's gripping force under an external magnetic field. **(h)** Relationship between gripping force and external magnetic field strength. **(i)** Relationship between compression resistance and external magnetic field strength.

Comment 3: In Section 2.1, the stability boundary model provided to determine bistability for samples with varying geometric parameters is well-executed, and the experimental results shown in Fig. 2(b) align nicely with predictions. To make this concept even more accessible, particularly for readers who may not specialize in mechanics, including a video showing the deformation process of the two types of samples would be highly beneficial.

Response 3: Thank you for your valuable suggestion. To provide a more intuitive understanding of the differences between bistable and monostable shells, we have included a set of comparative magnetic actuation experiments on shells of different thicknesses (radius $R = 5$ mm, thickness $h = 1.5$ mm for monostable case and $h = 0.5$ mm for bistable case) in the supplementary materials. **The results are presented in the newly added Supplementary Fig. S2 and the accompanying video (Supplementary Movie S1, also newly added).**

In the Text S1, we made the following revisions:

“To further illustrate the effect of shell thickness on its bistable properties, we present the experimental results shown in Fig. S2. Under an applied downward magnetic field of 65 mT, shells of varying thicknesses undergo deformation between the upward-convex (State A) and downward-concave (State B) configurations. In Fig. S2(a), for the shell with a thickness of $h = 1.5$ mm, the shell started to return to State A upon removal of the magnetic field at $t = 7.6$ s and failed to remain in State B, reverting to State A by $t = 15.9$ s, thus demonstrating monostable behavior. Conversely, Fig. S2(b) shows that the thinner shell ($h = 0.5$ mm) exhibits a different outcome: upon removal of the magnetic field at $t = 12.5$ s, the shell remained stable in State B throughout the observation period from $t = 12$ to 40 s), confirming its bistable characteristics (Supplementary Movie S1).”

Fig. S2. Deformation processes of monostable and bistable shells. (a) Monostable shell ($R = 5$ mm, $h = 1.5$ mm). (b) Bistable shell ($R = 5$ mm, $h = 0.5$ mm).

Comment 4: In Lines 139–147, the comparison of displacements between bistable and "conventional" actuators seems to overlook the design intent of conventional structures, which are meant to avoid large deflections. Instead, the focus should be on the advantages of bistable structures, such as their rapid configuration transitions and ability to maintain deformation under zero load. I recommend removing the current comparison and discussing these unique advantages in more depth.

Response 4: Thank you for your valuable feedback regarding the design of conventional structures. We would like to offer some of our perspectives on this issue. In conventional actuators, large deflections are indeed avoided because the internal elastic restoring force increases with the deformation and resists further shape transformation. As a result, in practical applications, these structures often rely on continuously increasing the magnetic field strength to achieve large shape transformation. However, **we think that "avoiding large deflections" is not the design intent of conventional structures, but rather a limitation inherent to their design.** Addressing this limitation and achieving significant shape transformation under low magnetic fields is a key objective in this field, including in our work. To highlight this, we included a comparison of displacements between bistable and "conventional" actuators in the previous manuscript, emphasizing the advantage of our hemispherical bistable actuators in this aspect—one of the main goals of our research. **Therefore, we have retained this comparison in the revised manuscript and hope for your understanding.**

We fully acknowledge your perspective and further emphasize the intrinsic advantages of bistable structures, such as their rapid configuration transitions and the ability to retain deformation under zero external load. In response, we have conducted further studies on the two characteristics under varying magnetic field conditions and included more detailed data to support our findings. On one hand, following the suggestions in Comment 2 and Comment 3, we investigated the deformation retention capability of hemispherical samples under different magnetic fields (Figure 6(i)) and with different shell thicknesses (Fig. S2), which is not reiterated here. On the other hand, to more clearly illustrate the configuration transition process, we updated Fig. 3(f) and

conducted additional experiments on the shell's transition capability under various magnetic fields and frequencies. We have also revised the relevant descriptions in the main text accordingly.

On page 7 (Section 2.1), we made revisions to the text:

“Furthermore, to evaluate the impact of the magnetic field on the configuration transition process, we applied direct current (DC) and alternating current (AC) magnetic fields with varying amplitudes and frequencies. Under a 60 mT DC field, the bistable actuator exhibited forward and reverse switching times of 0.243 s and 0.253 s, respectively (Fig. 3e, Supplementary Movie S4). Under an AC field, we measured and analyzed how the switching time varied with different magnetic field parameters (Fig. 3f and Supplementary Movie S5). At a fixed frequency, increasing magnetic field strength reduces the switching time. For instance, at 1 Hz, increasing the field strength from 40 mT to 80 mT reduces the switching time from 0.667 s to 0.356 s. Similarly, at a fixed field strength, increasing frequency shortens the switching time. For example, at 80 mT, the switching time decreases from 0.356 s to 0.07 s as the frequency increases from 1 Hz to 6 Hz. Additionally, switching stability depends on both the frequency and strength of the magnetic field. For example, a 50 mT field allows stable switching up to 2 Hz but fails above 3 Hz. Thus, by precisely controlling the magnetic field’s amplitude and frequency, the actuator's response can be effectively tuned.”

Fig. 3. Deformation and shape transformation of the proposed magnetoactive bistable hemispherical soft actuators. (f) Switching time curve of the hemispherical sample under AC magnetic fields with different frequencies and intensities.

Comment 5: In Section 2.4, while the structures discussed in this section are fascinating, referring to them as metamaterials requires more detailed explanation. What specific types of metamaterials do they represent—mechanical, electrical, or photonic—and what unique properties qualify them as such? If these classifications and justifications cannot be clearly articulated, I encourage the authors to use a more appropriate term to describe these structures.

Response 5: Thank you for your insightful comment. First, we sincerely apologize for the lack of clarity in referring to the structures discussed in Section 2.3 (**updated section number**). Based on existing literature and the characteristics of our actuators, we think the structures we have designed fall under **the category of magneto-mechanical metamaterials**. In the revised manuscript, we have provided references to support this classification. Accordingly, we have updated the corresponding description in the revised version. If the revised explanations still leave room for doubt, we are willing to adopt alternative terminology to ensure clarity.

On page 10 (Section 2.3), we made revisions to the text:

“In response, we proposed a magneto-mechanical metamaterial design featuring shape reprogrammability at the unit-cell level, achieved through an array of our magnetoactive bistable soft actuators. Such metamaterial typically offers unique advantages for shape reconfiguration and property tuning by applying an external magnetic field, enabling rapid, reversible, and untethered actuation^[51, 52].”

Comment 6: In Section 2.5, the geometry of the bar in the gripper likely influences its grasping capability. I strongly encourage the authors to explore this aspect further through simulations or experiments, or at least discuss it more comprehensively in the manuscript.

Response 6: Thank you for your thorough review and valuable suggestion. In the revised manuscript, **we have expanded our analysis to include a discussion on the effect of tentacle (bar in the gripper) length and stiffness on the gripper**. Our

experimental results confirm your insight, showing that the geometry of the bar indeed plays a significant role in its grasping performance. This enhanced analysis is crucial for a deeper understanding of the developed gripper robot and will aid in improving its overall performance. **The results are presented in the newly added Text S3 and Fig. S12.**

In the Text S3, we added the following descriptions:

“Text S3: Influence of gripper tentacle on gripping effectiveness

We fabricated molds for sea anemone-inspired grippers with different tentacle lengths ($L = 1, 3, 5, 7$ mm) using 3D printing. A composite of 50 wt% magnetic silicone (NdFeB + Ecoflex 00-20) was cast into the molds to form the grippers (shell radius $R = 15$ mm), as shown in Fig. S12(a). Using a permanent magnet (N35, $D = 50$ mm, $H = 20$ mm), we tested the grippers' ability to grasp cylindrical objects of varying radii. When the tentacle length is relatively short ($L \leq 1$ mm), the gripper can only effectively grasp large objects ($R \geq 8$ mm) and performs poorly with smaller objects. As the tentacle length increases, the gripper's success rate in grasping smaller objects improves significantly, achieving a broader grasping range for objects with radii between 1 and 10 mm. However, when the tentacle length becomes too long ($L = 7$ mm or more), interference between the tentacles during the grasping process reduces the gripper's success rate. Based on these observations, we selected a tentacle length of $L = 5$ mm as the optimal design parameter for the gripper.

In addition, we investigated the effect of tentacle stiffness on grasping performance while keeping the tentacle geometry and distribution density constant. We fabricated grippers with four different tentacle materials: pure silicone (Ecoflex 00-20), magnetic soft composite (50 wt% magnetic powder, base Ecoflex 00-20), and rigid PLA (Fig. S12(c)). Using a permanent magnet, we tested the grippers' ability to grasp objects of three different masses (1.5 g, 3.0 g, 4.5 g) and measured the magnetic field threshold for switching during the grasping process. As shown in Fig. S12(d), increasing tentacle

stiffness improves the gripper's success rate in grasping heavier objects. However, overly rigid tentacles (Gripper C) requires significantly higher magnetic field thresholds for actuation. The magnetic field threshold for Gripper C (48 mT) is 1.6 times that of Gripper B (30 mT). Therefore, using magnetic soft materials for the tentacles provides an optimal balance between performance and actuation efficiency, outperforming both pure soft and rigid materials.”

Fig. S12. Effects of tentacle length and stiffness on the grasping efficiency of the sea anemone-inspired gripper. (a) Experimental setup showing grippers with different tentacle lengths (L). **(b)** Grasping success rates of grippers with different tentacle lengths for cylindrical objects of varying radii ($R = 1–10$ mm, height $H = 10$ mm). **(c)** Experimental setup showing grippers with tentacles of varying stiffness. **(d)** Grasping success rates of grippers with tentacles of varying stiffness for objects of different mass (1.5 g, 3.0 g, 4.5 g).

Comment 7: Carefully proofread the manuscript to minimize typographical and grammatical errors. Ensure consistent symbol formatting throughout the paper. For instance, while symbols in the supplementary material are italicized, those in line 118 are not. Adhering to conventional formatting rules (e.g., Times New Roman for Latin letters and italicized Greek letters for scalars) will enhance readability. Introduce all abbreviations with their full names when they first appear in the text. Consider using "wt%" instead of "mass fraction" for clarity, specially in figures. Adjust the placement of the scale bar in Fig. 2(d) for better visual alignment.

Response 7: Thank you for your constructive feedback. We have carefully reviewed and revised the manuscript to address your comments as follows.

Issue 1: Ensure consistent symbol formatting throughout the paper. For instance, while symbols in the supplementary material are italicized, those in line 118 are not. Adhering to conventional formatting rules (e.g., *Times New Roman* for Latin letters and italicized Greek letters for scalars) will enhance readability.

Response: We apologize for the inconsistencies in symbol formatting. We have revised all symbols in the main text and supplementary materials to follow conventional formatting rules (*Times New Roman* for Latin letters and italicized Greek letters for scalars). Given the extensive changes, we have not detailed all modifications in this response letter, but the revisions are highlighted in the updated manuscript for your reference. We hope these revisions address your concerns.

Issue 2: Introduce all abbreviations with their full names when they first appear in the text.

Response: We apologize for any confusion caused by unclear abbreviations. In the revised manuscript, we have ensured that all abbreviations are introduced with their full names upon first mention.

Issue 3: Consider using "wt%" instead of "mass fraction" for clarity, especially in figures.

Response: We appreciate this valuable suggestion and have replaced "mass fraction" with "wt%" throughout the manuscript and figures. As an example of the revisions made, the following outlines the specific changes.

On page 5 (Section 2.1), we made revisions to the text:

“Based on these findings, we further investigated the magnetic field required for shape transformation—referred to as the switching magnetic field—in silicone-based actuators ($R = 10$ mm, $h = 0.5$ mm) with varying mass fractions of magnetic powder (Fig. 2d). The magnetic powder content influences the switching magnetic field by affecting the Young's modulus and residual magnetization strength. Experimental results show that, in the given tests, the minimum switching magnetic field was 20 mT when the magnetic powder content was 50 wt%. Thus we selected a combination of 50 wt% magnetic powder and silicon substrate material for subsequent experiments. Additionally, we tested the variation in the switching magnetic field with different shell thicknesses and radii. As shown in Fig. 2e, the switching magnetic field increased with shell thickness but decreased with shell radius.”

Issue 4: Adjust the placement of the scale bar in Fig. 2(d) for better visual alignment.

Response: We apologize for the oversight in figure alignment. The scale bar in Fig. 2(d) has been repositioned for improved visual alignment.

Fig. 2. Preparation process and bistable behavior of the proposed magnetoactive bistable hemispherical soft actuators. (d) Experiment on the switching magnetic field of actuators with varying magnetic powder content (actuator dimensions $R = 10$ mm, $h = 0.5$ mm). Scale bar: 20 mm.

Response to the Reviewer #2:

General comment: The manuscript title “Magnetoactive Bistable Soft Actuators for Programmable Large Shape Transformations at Low Magnetic Fields by Quanliang Cao et al. introduces magnetoactive actuators made from silicon-based elastomers shaped in a dome structure. It demonstrates the potential for various applications, such as soft pumps, metamaterials, and soft grippers, due to its simple operating mechanism and fabrication process. Furthermore, the ability to achieve large deformations under low magnetic fields (and thus low energy consumption) appears to be a notable achievement in the field of magnetic actuators. However, since the publication of magnetic actuators by Metin Sitti's group[1], many studies on magnetic actuators have been conducted. This raises the question of whether simply creating a dome-shaped actuator truly adds novelty to the field. Additionally, creating soft pumps, metamaterials, and grippers is not fundamentally a new concept. There are also several concerns and questions regarding the manuscript. In the current stage, this manuscript is therefore not suitable for publication in Nature communications.

- *Ref. [1]: Hu, W., Lum, G., Mastrangeli, M. et al. Small-scale soft-bodied robot with multimodal locomotion. Nature 554, 81–85 (2018).*

Response: Thank you for your thoughtful feedback on our manuscript. We appreciate your concerns and would like to offer some clarifications.

Regarding the contribution of our work to the field, we recognize that magnetic soft actuators have become an important area of research in recent years. While much research has been conducted on magnetic actuators, the novelty of our work lies in the integration of hemispherical geometry with bistable magnetic actuation. This unique combination enables large shape transformations under low magnetic fields and the ability to maintain stable concave or convex states without external magnetic fields, while also allowing for stiffness modulation in the presence of a magnetic field. Therefore, **our design provides a simple, energy-efficient solution with high**

programmability to overcome the challenges faced by existing magnetic soft actuators, particularly in terms of shape control. We believe this approach **could pave the way for further exploration of bistable or multistable structures in magnetic soft actuators**, thereby injecting new vitality into the field. This point has been emphasized in the revised Discussion section of the manuscript.

Regarding the novelty of the applications, we acknowledge that creating soft pumps, metamaterials, and grippers is not a fundamentally new concept and has indeed received significant attention in previous research. Precisely because of this, **we chose these typical applications to demonstrate the effectiveness and advancement of our approach**. Importantly, we would like to emphasize that **our true innovation lies in the unique actuator structure and its enhanced functionality**, which distinguishes our work from prior research. This distinction has been clearly addressed in the context of applied research, where we explain both the innovative design and its differences from existing work. We hope this clarifies our intention.

For other concerns you raised, we will provide a point-by-point response in the following sections.

Comment 1: In the overall structure, the magnetoactive bistable soft actuators are made with a 50% magnetic particle composition of Ecoflex 00-20 and a thickness of 0.5 mm. However, the state A without an applied magnetic field is not presented anywhere. Is this because the structure cannot maintain itself in that state? If so, PDMS might be a more favorable option.

Response 1: Thank you for your comment, and we apologize for any misunderstanding caused by unclear expressions in the manuscript. Regarding your concern, in fact, **both state A and state B can maintain their shape stability in the absence of an external magnetic field**. This also highlights one of the key features of our bistable actuator. To support this point, **we have added Figs. S3–S4 and Supplementary Movie S2 in the revised manuscript**. Additionally, we have revised the corresponding text for better clarity.

On page 5 (Section 2.1), we made revisions to the text:

“As shown in Fig. 2b, the hemispherical soft actuators can exhibit either bistable or non-bistable (monostable) characteristics under different structural parameters. Specifically, as the ratio of the radius to thickness increases, bistability becomes more likely, which is consistent with the bistable boundary model presented in *Supporting Information*, Text S1. As an example, the deformation processes of monostable and bistable shells are provided in *Supporting Information*, Fig. S2, and Supplementary Movie S1. Under bistable conditions, both concave-up and convex-down shapes can maintain their stability in the absence of an external magnetic field, as shown in *Supporting Information*, Figs. S3-S4, and Supplementary Movie S2.”

In the revised supplementary Text S1, we added the following changes:

“Furthermore, we experimentally demonstrate that, under bistable conditions, both concave-up and convex-down shapes can maintain their stability in the absence of an external magnetic field. Taking a magnetic shell with $R = 10$ mm, $h = 0.5$ mm, and 50 wt% magnetic powder as an example, we set the initial state to state B (Fig. S3). From 0 to 7 s, the shell remained stable in state B without an external magnetic field. At $t = 7$ s, we applied a 20 mT upward vertical magnetic field by powering a DC current source (current set to 3 A). The shell had begun transitioning from state B to state A. By $t = 9.7$ s, the shell fully switched to state A. At $t = 15.9$ s, the DC power was turned off, and the shell remained stable in state A without a magnetic field from $t = 15.9$ to 25.4 s. Similarly, we set the shell's initial state to state A, reversed the connection of the DC source and the coil to apply a downward magnetic field, and repeated the process. The results are shown in Fig. S4. These two sets of experiments clearly demonstrate that the shell can remain stable in both state A and state B without an external magnetic field, and the dynamic switching between the two states is achieved by controlling the applied magnetic field.”

It is noted that the switching process took a relatively long time (approximately 2.7 s and 5 s). This is because the switching magnetic field intensity was set at 20 mT, close to the critical threshold shown in Fig. 2(e). Increasing the magnetic field intensity enhances the magnetic torque on the shell, significantly reducing the switching time between the two stable states.

Fig. S3. Bistable dynamic process showing the transition from State B to State A.

Fig. S4. Bistable dynamic process showing the transition from State A to State B.

Comment 2: The design and principle of the soft pump are strikingly similar to those in ref [2]. Additionally, the results in that reference are similar to, or superior to, those in this study in terms of the time required for liquid transport, the range of minimum magnetic fields, and the conversion frequency.

- *Ref. [2]: Lin D, Yang F, Gong D, et al. Bio-inspired magnetic-driven folded diaphragm for biomimetic robot [J]. Nature Communications, 2023, 14(1): 163.*

Response 2: Thank you for your valuable comment. We acknowledge the impressive design and performance of the soft pump proposed by Prof. Fan Yang's group are indeed impressive. They introduced a novel soft actuator with a folded magnetic diaphragm for pumping at low magnetic fields. Their work on pump system design and performance analysis has provided inspiration for our research. However, there are important differences between our work and theirs in terms of actuator structure, driving magnetic field waveform, and valve design. Upon your reminder, we realized that although we referenced their work in the original manuscript, it was not sufficiently highlighted. **Therefore, we have added the relevant content in the revised manuscript to more clearly highlight these differences and better contextualize Prof. Fan Yang's work in relation to our own.**

The added descriptions of Prof. Fan Yang 's work (Ref. [15] in the revised manuscript) are summarized as follows.

On page 7 (Section 2.2):

“Recent advancements have shifted focus towards soft pumps driven by alternative sources such as electricity ^[40, 41], heat ^[42, 43], and magnetic fields ^[15, 44-46], enhancing the field and expanding its potential applications.”

On page 7-8 (Section 2.2):

“As shown in Fig. 4a, under the influence of an alternating magnetic field, the shape change of the soft actuator leads to continuous changes in the chamber volume, while the soft valves on both sides alternately open and close, thereby achieving the pumping action. The composition of this pump and pumping principle are similar to the soft pump system with a folded magnetic diaphragm proposed by Lin et al. ^[15], but there are differences in the structure of the magnetic actuator, the magnetic-induced deformation mechanism, and the soft valve design. In this application, the pump's valves are composed of magnetic soft materials embedded with NdFeB particles, and the magnetization design of the left and right valves is shown in Fig. 4b, resembling the

valve design from our previous work on magnetically controlled capsules ^[10]. Unlike single-layer magnetic soft valves ^[15], our double-layer design enables the valves to self-close in the absence of an external magnetic field, driven by the attractive gradient magnetic force between the valve frame and the leaf. This mechanism helps prevent potential liquid backflow after the magnetic field is removed.”

On page 9 (Section 2.2):

“Meanwhile, importantly, like the work of Lin et al. ^[15], our approach is also impressive in significantly reducing the driving magnetic field required for pumping (Supporting Information, Table S1).”

Regarding your concerns about the soft pump’s performance in terms of the “time required for liquid transport, the range of minimum magnetic fields, and the conversion frequency,” we believe that both technical approaches have shown good performance in this application direction. Both approaches can operate across a relatively wide range of conversion frequencies and help reduce the magnetic field amplitude. However, our work has an advantage in reducing the time required for liquid transport (i.e., increasing the liquid transport velocity), as clearly demonstrated in *Supporting Information*, Table S1. **Notably, in response to your feedback, we have supplemented the data in the revised manuscript to better support the above conclusions.** One important addition is the inclusion of experiments with our bistable pumps driven by a sinusoidal magnetic field of 10 Hz and 20 mT, where this frequency exceeds the highest liquid pumping frequency (3 Hz) reported in our previous manuscript. This demonstrates that our magnetoactive soft pump can achieve effective pumping at high frequencies (*Supporting Information*, Fig. S8, and Supplementary Movie S8, newly added).

On page 9 (Section 2.2), we added the following descriptions to the text:

“Compared to similar existing magnetoactive soft diaphragm pumps in the *Supporting Information*, Table S1, and other miniaturized soft pumps described in Zhou et al.’s

work [47], our developed soft pump (> 30 mL/min) ranks among the top in terms of liquid output flow rates. Meanwhile, importantly, like the work of Lin et al. [15], our approach is also impressive in significantly reducing the driving magnetic field required for pumping (*Supporting Information*, Table S1). It should be noted that our magnetoactive soft pump can achieve effective pumping even at higher frequencies than those shown in Fig. 4g (e.g., at 10 Hz and 20 mT sinusoidal field, with a liquid pumping rate of 10.5 mL/min, as shown in *Supporting Information*, Fig. S8, and Supplementary Movie S8). However, due to limitations of the current power supply, the achievable magnetic field amplitude is restricted, and thus, a more in-depth study of pumping performance at these higher frequencies is not conducted in this work."

Fig. S8. Liquid pumping process under a 10 Hz, 20 mT sinusoidal magnetic field.

Comment 3: The papers cited as references for metamaterials focus on studies with a single unit cell [3]. However, the experiments in this paper are more akin to a 'meta-surface'. When considering the research scope of meta-surfaces, the application of magnetoactive bistable soft actuators is somewhat lacking in terms of lower resolution and binary stage changes (0 and 1) [4].

- Ref [3]: Jiao, P., Mueller, J., Raney, J.R. et al. *Mechanical metamaterials and beyond. Nat Commun 14, 6004 (2023).*
- Ref [4]: Johnson, B.K., Naris, M., Sundaram, V. et al. *A multifunctional soft robotic shape display with high-speed actuation, sensing, and control. Nat Commun 14, 4516 (2023).*

Response 3: Thank you for your valuable feedback. As you pointed out, the magnetoactive metamaterials discussed in Section 2.3 indeed share similarities with existing research on meta-surfaces. Metasurfaces are typically considered "planar metamaterials" or "2D metamaterials" (*Rep. Prog. Phys.* 2016, 79, 076401; *Proceed. IEEE* 2022, 110, 31–55). However, it is important to note that our actuator utilizes a three-dimensional hemispherical structure, which distinguishes it from meta-surface materials in terms of both design and functionality.

Regarding the resolution and binary stage changes, the continuous-type materials in the reference you mentioned and the binary-type materials used in our study represent two common forms or directions in the fields of display and storage, each with its own advantages and disadvantages. Continuous-type materials excel in precise control and smooth deformation but are more complex in terms of control and fabrication. On the other hand, binary-type materials are characterized by simple switching features, higher stability, and lower control complexity. **We have emphasized this point in Section 2.3 of the revised manuscript.** Additionally, as mentioned in the discussion section of the manuscript, looking ahead, with the application of advanced manufacturing techniques such as 3D printing, we plan to further scale down the bistable soft actuators to submillimeter dimensions, enhancing their potential for use in high-resolution systems.

On page 10 (Section 2.3), we made revisions to the text:

“Compared to the continuous-type shape display ^[53, 54], the binary-type presented in this work, while less effective at achieving smooth deformation control, stands out for its straightforward switching mechanism, superior stability, and reduced control complexity. Furthermore, although our metamaterial operates in a manner similar to the approach by Chen et al. ^[30], there are significant differences in the magnetic structure design and actuation methods. Our work integrates magnetic components with mechanical motion components through magnetic soft materials, offering higher integration and simpler structures, which could provide a new pathway for achieving higher levels of dynamic control over the mechanical properties of metamaterials.”

Comment 4: There are several issues with numbering and missing sections throughout the manuscript. For example, the order of Fig. 3b and 3c does not match the sequence in the manuscript, and there is no Results 2.3.

Response 4: Thank you for your feedback. We sincerely apologize for the oversight regarding the numbering and missing sections. We appreciate your attention to these details. In the revised manuscript, we have thoroughly reviewed the entire text and made the necessary corrections, including fixing the order of Figs. 3b and 3c, and rechecking the chapter numbering. **The revised framework of section "2. Results" and Figure 3 are shown below.**

2.1 Design and performance of magnetoactive bistable soft actuator

2.2 Magnetoactive soft pumps

2.3 Reprogrammable magnetoactive metamaterials

2.4 Magnetoactive biomimetic soft grippers

Fig. 3. Deformation and shape transformation of the proposed magnetoactive bistable hemispherical soft actuators.

Comment 5: Furthermore, the manuscript repeatedly claims higher performance compared to previous work, but it is necessary to clearly state these comparisons with existing studies.

Response 5: Thank you for your constructive feedback. We apologize for any lack of clarity in our comparisons with previous work. **To address this, we have revised the relevant descriptions to ensure that our results are explicitly compared with existing studies.** We think that, with your valuable suggestions, the presentation and comparison of the performance of our actuator are now clearer.

On page 6 (Section 2.1), we made revisions to the text:

“For the comparison, both the bistable actuator and the conventional circular actuator are made from a magnetic soft composite material containing 50 wt% NdFeB, with identical planar geometric parameters (radius $R = 20$ mm, height $h = 1$ mm). The bistable actuator is magnetized axially, while the circular actuator undergoes pre-deformation using a convex mold with a height equal to R before being magnetized axially (see *Supporting Information*, Fig. S5 for detailed geometry). Upon exposing both actuators to external magnetic fields (Figs. 3a-3c), we observe that, compared to the conventional circular actuator, the bistable actuator exhibits distinct deformation characteristics and demonstrates superior deformation capabilities at low magnetic fields. Specifically, unlike the circular actuator, which shows a gradual increase in deformation with increasing magnetic field, the bistable actuator displays a pronounced discontinuous transition around 40 mT, with minimal shape change before and after this threshold (Fig. 3a). This bistable behavior is exhibited as a sudden deformation, either from convex to concave or vice versa, showing a certain degree of symmetry (Fig. 3b). Notably, this abrupt deformation mode allows the bistable actuator to achieve a significantly larger shape change at lower magnetic fields. For example, at 40 mT, the bistable actuator reaches a positive height exceeding 19 mm, while the conventional circular magnetoactive actuator deforms by less than 5 mm (Fig. 3c). Even when the magnetic field is increased by an order of magnitude, the deformation of the conventional actuator still fails to match that of the bistable actuator.”

On page 9 (Section 2.2), we made revisions to the text:

“Compared to similar existing magnetoactive soft diaphragm pumps in the *Supporting Information*, Table S1, and other miniaturized soft pumps described in Zhou et al.'s work ^[47], our developed soft pump (> 30 mL/min) ranks among the top in terms of liquid output flow rates. Meanwhile, importantly, like the work of Lin et al. ^[15], our approach is also impressive in significantly reducing the driving magnetic field required for pumping (*Supporting Information*, Table S1).”

On page 14 (Section 3), we made revisions to the text:

“In comparison to existing bistable dome-shaped soft actuators (*Supporting Information*, Table S2), our design offers notable advantages in reducing state-switching time (< 0.1 s, dynamically tunable) and lowering the required driving magnetic field (< 100 mT).”

Comment 6: While the comparison in the Supplementary Information is more organized than expected, it lacks visibility, and some points are inaccurate (e.g., in the data from reference 9 in the supplementary information, when both membranes are applied, the average flow rate is higher than 7 mL/min).

- *Ref. [9]: Lin D, Yang F, Gong D, et al. Bio-inspired magnetic-driven folded diaphragm for biomimetic robot [J]. Nature Communications, 2023, 14(1): 163.*

Response 6: Thank you for your insightful comments. **We have significantly revised Table S1 in the Supporting Information to address your concerns**, including streamlining the comparison targets and content, providing additional details (such as material types and driving magnetic field frequencies), and carefully correcting and updating the data. Specifically, we made the following main changes:

1. Table Refinement: We have focused the comparisons in Table S1 on magnetic soft diaphragm pumps most relevant to our work, removing less pertinent studies. Meanwhile, to further enhance clarity, we have eliminated the "Pressure" and "Applications" columns, which contained less critical information. This adjustment makes the table more concise, improving both its readability and clarity.

2. Magnetic Parameter Section: In response to your comment, we have updated the magnetic field details in Ref. [9] (Ref. [11] in the new Supporting Information), including information for both the single diaphragm pump and the double diaphragm pump. For the single diaphragm pump, the driving magnetic field was listed as a sinusoidal field with a frequency of 2 Hz and an amplitude of 40 mT. For the double diaphragm pump, the driving magnetic field was listed as a sinusoidal field with a frequency of 5 Hz and an amplitude of 20 mT. These data are consistent with the magnetic field information provided in the supplementary video of the reference (Movie S1).

3. Average Liquid Flow Rate Section: To ensure the credibility of the flow rate data in Ref. [9] (Ref. [11] in the new Supporting Information), **we performed our calculations based on the data provided in the supplementary video of the reference (Movie S1)**, as we believe this method yields more accurate results. Specifically, for the single diaphragm pump, the initial liquid level at 0 s was 1.05 mL, and by the end of the pumping process (8.01 s), it reached 2.10 mL. Therefore, the average output flow rate was calculated as 7.9 mL/min. For the double diaphragm pump, the initial liquid level at 0 s was 0.95 mL, and by the end of the pumping process (6.54 s), it reached 1.80 mL. Consequently, the average output flow rate was calculated as 7.8 mL/min.

4. Corrections to Parameters in Our Work: To better compare and highlight the advantages of our approach, we provided results under two experimental conditions: the average flow rates obtained at an equivalent magnetic field of 36.75 mT for frequencies of 2 Hz and 5 Hz, in which the values of magnetic field frequency align with the corresponding values in Ref. [9] (Ref. [11] in the new *Supporting Information*). Additionally, we clarified the relationship between the equivalent magnetic field value and the field amplitude in the table.

We believe these revisions address the issues raised and improve the overall presentation of the data in the Supporting Information. Thank you again for your constructive feedback.

The updated Table S1 in the Supporting Information is shown below.

Table S1. Performance comparison of magnetoactive soft diaphragm pumps.

Reference	Actuation type	Material type	Magnetic parameter	Size (mm)	Average Liquid flow rate (mL/min)
[7]	Magnetic gradient force	TPR (50A shore hardness) + Carbonyl iron particles	Coil: 443 mT, 0.5 Hz	$\Phi \approx 40^{\#}$	16.2
[8]	Magnetic gradient force	PDMS + Carbonyl iron particles	Coil: 225mT, 1 Hz	30×10×0.7	1.974
[9]	Magnetic gradient force	Prepolymer (Neukasil RTV 23) + Iron particles	Coil: 430 mT, 1 Hz	$\Phi 12 \times 320$	78.5*
[10]	Magnetic torque	Flexible photosensitive resin (F39T) + NdFeB particles	Permanent magnet: periodic motion	$\Phi 44 \times 1.8$	3.6
[11]	Magnetic torque	Silicone Elastomer (Ecoflex00-20)+ NdFeB particles	Coil: 40 mT, 2Hz (Single diaphragm)	$\Phi 20 \times 0.5$	7.9
			Coil: 20 mT, 5Hz (Double diaphragm)		7.8
Our work	Magnetic torque	Silicone Elastomer (Ecoflex00-20)+ NdFeB particles	Coil ⁽¹⁾ : 36.75 mT, 2Hz	$\Phi 40 \times 1$	22.72
			Coil ⁽¹⁾ : 36.75 mT, 5Hz		30.1

#: This value is not provided directly in the original article but is obtained through our estimation.

*: This represents the maximum pumping flow rate when the pump body is operating. Since the article does not provide additional data, it cannot be converted into an average flow rate.

Coil⁽¹⁾: For the energy-saving waveform with a frequency of 2Hz, a magnetic field of 50mT, and a duty cycle of 0.2 used in our work, the corresponding equivalent amplitude of a sinusoidal magnetic field with the same energy consumption is 36.75mT (calculated using Eq. (9) in Text S2).

TPR: Thermoplastic rubber materials. NdFeB: Neodymium-iron-boron. PDMS: polydimethylsiloxane.

Red font: This indicates that the flow rate is relatively low (< 20 mL/min), or that the applied driving magnetic field is relatively high (> 100 mT).

Response to the Reviewer #3:

General comment: I co-reviewed this manuscript with one of the reviewers who provided the listed reports. This is part of the Nature Communications initiative to facilitate training in peer review and to provide appropriate recognition for Early Career Researchers who co-review manuscripts.

Response: Thank you for your feedback and for taking the time to co-review our manuscript. We have made the necessary revisions based on the reviewers' comments and have provided detailed responses to each of them in the above part.

=====

Response to the Reviewer #4:

=====

General Comment: The authors demonstrated a magnetically driven soft bistable actuator that can maintain two stable states in the absence of a magnetic field and achieve shape changes in a low-intensity magnetic field. In addition, the authors demonstrated the applications of the developed actuator for pumping liquids or gases, programmable metamaterials for information encoding and soft grippers. However, the core concept of combining magnetic actuation with bistable structures has been extensively explored in prior literature. While the authors claim improvements in switching time and reduced magnetic field requirements (<100 mT), these metrics are not groundbreaking compared to recent literature. For example, similar performance has been achieved with other bistable designs (Pal & Sitti, PNAS, 120, e2212489120 (2023); Tang et. al, Sci. Robot. 9, eadm8484 (2024)). In addition, the theory and deformation of bistable spherical shells have been extensively studied (e.g. Sci. Robot. 5, eabb1967 (2020)). The demonstrated applications (soft pumps, metamaterials, grippers) are conceptually similar to existing systems. While the study is technically sound, it looks like lack of enough novelty and contribution to the advancement of this field, it does not meet Nature Communications's threshold for novelty and broad impact. The work appears incremental, with insufficient differentiation from prior art in bistable magnetic actuators. The research topic and approach are good, but scientific findings and technical improvement are required for the broad readership of the journal. Below are some comments on the manuscript for the author's reference.

Response: We sincerely thank you for the thorough evaluation of our manuscript and for acknowledging the technical soundness of our work. We value the opportunity to further clarify the unique contributions of our study and have made improvements accordingly. Below, we provide our detailed responses to the general remarks:

1. Clarification of Innovation and Distinction from Prior Works

We fully agree with your comment that the combination of magnetic actuation and bistable structures is not entirely novel, which we have explicitly acknowledged

in the literature review of the original manuscript. In our previous submission, we recognize that the unique features and distinctions of our work were not sufficiently articulated, resulting in an inadequate presentation of its innovative aspects. Importantly, **following your valuable suggestion, we have now clearly defined the challenges related to achieving large deformations under low magnetic fields, especially under fully constrained boundary conditions.** This refinement significantly helps to highlight the advantages and innovations of our work.

In fact, integrating bistable mechanisms with different magnetically actuated structures can lead to significantly varied functional behaviors. Previous studies differ considerably in their structural focus and the specific challenges they address. For clarification, we distinguish our work from the two representative studies mentioned (noting that Gorissen et al. (Sci. Robot. 2020) employed pneumatic actuation under unconstrained conditions):

- Pal and Sitti (PNAS, 2023) focused on the programmable control of arrays of bistable beam structures, employing magnetic fields to switch planar bistable beams and developing customized logic for control.
- Tang et al. (Sci. Robot., 2024) aimed to address the challenges of slow energy accumulation and limited jump height in soft robots by designing a pyramid-shaped magnetic bistable jumping robot capable of millisecond-scale jumping through high-energy release. **(A detailed description of this work has been included in the revised manuscript)**

These two works differ substantially from ours in terms of actuator structure, boundary conditions, and application objectives. As clarified in the manuscript, our study proposes a magnetically actuated bistable soft actuator based on a hemispherical shell structure, capable of achieving large-scale shape transformations under low magnetic fields. The deformation mode enabled in our system—shell deformation under strict edge confinement—is **fundamentally different from the two-end-constrained beam structure** in Pal & Sitti, the **unconstrained pyramid structure** in Tang et al., and the **unconstrained spherical shell structure** in Gorissen et al. It should be noted that the work by Gorissen et al. is pneumatically actuated rather than magnetically driven and involves an unconstrained configuration. While

their studies on the theory and deformation process provide some foundational support for our work, significant differences remain between their system and ours. To further clarify the research problem, **we have explicitly emphasized the role of constrained boundary conditions in the revised manuscript.**

Therefore, the scientific question and challenge addressed in our study are distinct and independent from those in previous work. Our approach and solution provide a differentiated contribution by **introducing the bistable magnetic actuation mechanism into a previously unresolved problem space**, which is achieving large-scale shape transformation under low magnetic fields and strict constraints in magnetically actuated soft actuators.

The relevant above discussions have been incorporated into the *Introduction* and *Discussion* sections of the revised manuscript.

2. Clarification on Performance Metrics and Technical Advancements

We sincerely appreciate your insightful suggestion regarding the definitions of “low magnetic field” and “large deformation”, which has helped to clarify the objectives and significance of our work. In the revised manuscript (see **the second paragraph of Introduction and Table S1**), we have provided clear and quantitative definitions for these key terms:

- A low magnetic field is defined as $B < 100$ mT.
- A large deformation is defined as a shape change ratio greater than 0.5, where the shape change ratio is calculated as defined as the ratio of the actuator’s maximum deformation to a characteristic structural dimension (e.g., diameter or length).

Under these clearly defined criteria, our work directly addresses the core question of how to achieve large-scale shape transformation (shape change ratio > 0.5) under low magnetic fields ($B < 100$ mT) and strict boundary conditions. Specifically, we demonstrate **a shape change ratio exceeding 0.8 under magnetic fields below 50 mT**, which **significantly outperforms existing magnetoactive actuators** operating under similar conditions.

Therefore, the significance of our contribution lies not only in the improvement of switching speed and reduction of actuation field intensity, but more importantly, in

offering an effective solution to the challenge of realizing large-scale shape transformation in magnetically actuated soft systems under low magnetic fields and strict constraints.

3. Clarification on the Concern Regarding Demonstrated Applications

While soft pumps, metamaterials, and soft grippers are indeed common application forms in the field of soft actuators, we would like to emphasize that the core actuation structure proposed in our work—a magnetoactive bistable spherical-shell soft actuator—is distinct from those employed in previous studies (e.g., Lin et al., Nat. Commun., 2023; Chen et al., Nature, 2021). **These structural differences have a direct impact on both the design of the magnetic actuation and the resulting system-level performance.** For instance, in the soft pump application, we leveraged the characteristics of our actuator to **develop an energy-efficient, non-conventional sinusoidal magnetic field waveform.** Compared to existing magnetoactive soft diaphragm pumps, our device achieves a liquid output flow rate exceeding 30 mL/min, **ranking among the top level reported to date, and leads in pumping efficiency per unit magnetic field.** Moreover, it enables precise closed-loop flow control, a feature rarely realized in comparable systems.

Furthermore, through the implementation of three representative applications, we demonstrate not only the fundamental advantage of achieving large deformation under low magnetic fields, but also several distinct functionalities: fast actuation response (validated via the soft pump); bistable state programmability and structural simplicity (demonstrated in programmable metamaterials); and magnetically tunable stiffness with adaptive grasping capabilities (realized in soft grippers). **These features are further emphasized in the *Discussion* section of the revised manuscript.**

Therefore, **although the application categories may appear conceptually similar to existing works, the functional outcomes, performance characteristics, and implementation mechanisms are different.** These results collectively validate the novelty and practical utility of the proposed actuation mechanism.

In the second paragraph of *Introduction* section, we made the following revisions:

To address the aforementioned limitations, integrating bistable structures into magnetically actuated systems may provide a promising breakthrough. Bistability, which refers to the ability of a structure to maintain two stable states after experiencing mechanical instability, has emerged as a powerful strategy to enhance the performance and functionality of soft actuators [20, 21]. By utilizing bistable mechanisms, it is possible to achieve force amplification, high-speed movements, and shape retention without continuous-applied external actuation [22, 23]. When integrated with magnetic materials and corresponding magnetic actuation strategies, these bistable elements offer a synergistic combination of contactless actuation and mechanical stability, further improving system reliability, energy efficiency, and controllability. A representative example is provided by Tang et al. [24], who skillfully harnessed the rapid response and high energy release of a bistable mechanism to develop a pyramid-shaped, magnetically actuated jumping robot. This system demonstrated millisecond-scale actuation and achieved an impressive self-propulsion height that exceeded 100 times its own body length, which highlights its exceptional energy conversion efficiency under low magnetic fields. This emerging field has seen increasing research activity in recent years, resulting in the creation of diverse magnetoactive bistable structures, including beam-shaped [25-27], dome-shaped [28-31], origami and kirigami structures [24, 32-34], spanning from one-dimensional (1D) to three-dimensional (3D) configurations. Among them, dome-shaped configurations, as a representative 3D bistable architecture, could stand out for their significant application potential in magnetoactive soft actuators, owing to their structural simplicity and their ability to achieve more substantial volumetric transformations than planar structures [35]. However, despite their structural advantages, current research on dome-shaped magnetoactive bistable soft actuators has paid limited attention to improving their shape-switching performance. In particular, under practical conditions involving fully edge-constrained conditions, their deformation capability remains significantly restricted, hindering the full exploitation of the bistable mechanism. As a result, achieving large-scale shape transformations under low magnetic fields ($B < 100$ mT), specifically a shape change ratio exceeding 0.5,

where the ratio refers to the actuator's maximum deformation divided by a characteristic structural dimension (e.g., diameter or length), remains a major challenge (see *Supporting Information*, Text S1 and Table S1). This limitation constrains the broader deployment and functional advancement of magnetoactive soft actuators in complex real-world applications.

In the first paragraph of *Discussion* section, we made the following revisions:

In this work, we present a unique magnetoactive actuator that integrates magnetic actuation, soft materials, and bistable mechanics within a three-dimensional hemispherical shell embedded with NdFeB microparticles. This design overcomes the strong correlation between deformation amplitude and the applied magnetic field strength, a limitation commonly observed in conventional magnetoactive soft actuators. The actuator enables efficient switching between bistable states under low-strength pulsed magnetic torque disturbances, while remaining stable in the absence of an external magnetic field. This advancement addresses the persistent challenge of relying on continuous and relatively high magnetic fields (>100 mT) to sustain large-scale shape transformations, particularly those with shape change ratios exceeding 0.5 and occurring under constrained boundary conditions. Compared to existing magnetoactive soft actuators (*Supporting Information*, Tables S1 and S3), our design significantly reduces the state-switching time to below 0.1 seconds with dynamic tunability, and achieves large-scale shape transformations with a shape change ratio exceeding 0.8 under magnetic fields below 50 mT, thereby significantly surpassing previously reported performance benchmarks in this field. Furthermore, we demonstrated the broad applicability and functional diversity of our magnetoactive bistable actuator through three representative applications: a high-efficiency soft pump, a reprogrammable metamaterial, and a variable-stiffness soft gripper. These applications not only validate the actuator's capability for large, reversible deformations under low magnetic fields, but also highlight its potential in enabling closed-loop fluid control, logic-embedded 3D architectures, and adaptive, stiffness-tunable gripping.

Comment 1: Can the authors provide a more detailed comparison of the quantitative improvement in the driving magnetic field, and do more scientific parameterized comparison (Table S1)?

Response 1: We sincerely thank you for this valuable suggestion. In response, **we have substantially revised Table S2** (previously Table S1, updated due to reordering) to provide a more scientific and quantitative parameterized comparison between our actuator and representative existing magnetoactive soft diaphragm pumps. Specifically, **three additional data columns have been added** for comparison in the revised manuscript. The key improvements are as follows:

(1) Classification of actuation mechanisms based on magnetic field requirements:

A new column has been added to indicate whether each system relies on a magnetic field gradient. Based on their actuation mechanisms, existing magnetoactive soft pumps can be categorized into two types:

- Force-driven pumps, which require both magnetic field strength and a spatial field gradient for actuation;
- Torque-driven pumps, which operate under uniform magnetic fields without the need for a gradient.

This classification clarifies the varying levels of complexity in magnetic field design across different systems.

(2) Inclusion of normalized performance metrics for a more rigorous and quantitative comparison of magnetic field efficiency:

Two new normalized metrics that are normalized by magnetic input have been introduced:

- **Normalized Shape Change Ratio (mT^{-1}):** This newly added metric is calculated by dividing the shape change ratio—defined as the ratio of the actuator's maximum displacement to a characteristic structural dimension (e.g., diameter or length)—by the applied magnetic field strength. It reflects the actuator's deformation capability per unit magnetic field strength under practical operating conditions.

- **Normalized Flow Rate (mL/min/mT):** This newly added metric quantifies the fluid output per unit magnetic field strength, offering a direct evaluation of pump performance relative to magnetic input.

Through this comparison, it is evident that our actuator demonstrates leading performance among similar devices in terms of both normalized metrics. We believe that, following your valuable suggestions, the revisions have made Table S2 substantially more robust and transparent, more clearly highlighting the quantitative advantages of our actuator in achieving low-field, high-efficiency operation. We hope these improvements satisfactorily address your concerns regarding the scientific rigor and comparability of the presented data.

In the last paragraph of Section 2.2, we made the following revisions:

Notably, our approach also achieves a substantial reduction in the required driving magnetic field, resulting in a leading normalized shape change ratio during pumping and a superior flow rate per unit magnetic field compared to existing works (*Supporting Information*, Table S2).

The revised Table S2 in the *Supporting Information* is shown below.

Table S2. Performance comparison of magnetoactive soft diaphragm pumps.

Reference	Actuation Principle	Gradient-Free Actuation	Material type	Magnetic parameter	Normalized shape change ratio (mT ⁻¹)	Size (mm)	Average flow rate (mL/min)	Normalized flow rate (mL/min/mT)
[13]	Magnetic force	×	TPR (50A shore hardness) + Carbonyl iron particles	Coil: 443 mT, 0.5 Hz	$< 8 \times 10^{-5\#}$	40×40×18 [#]	16.2	3.65×10^{-2}
[14]	Magnetic force	×	PDMS + Carbonyl iron particles	Coil: 225mT, 1 Hz	2.1×10^{-3}	30×10×0.7	1.974	8.77×10^{-3}
[15]	Magnetic force	×	Prepolymer (Neukasil RTV 23) + Iron particles	Coil: 430 mT, 1 Hz	$< 1.5 \times 10^{-3\#}$	Φ12×320	78.5*	0.18*
[16]	Magnetic torque	√	Flexible photosensitive resin (F39T) + NdFeB particles	Permanent magnet: periodic motion 0.14 Hz**	—	Φ44×1.8	3.6	—
[7]	Magnetic torque	√	Silicone Elastomer (Ecoflex00-20)+ NdFeB particles	Coil: 40 mT, 2Hz (Single diaphragm)	$< 2 \times 10^{-3\#}$	Φ20×0.5	7.9	0.20
				Coil: 20 mT, 5Hz (Double diaphragm)	$< 3.5 \times 10^{-3\#}$		7.8	0.39
Our work	Magnetic torque	√	Silicone Elastomer (Ecoflex00-20)+ NdFeB particles	Coil ⁽¹⁾ : 36.75 mT, 2Hz	6.8×10^{-3}	Φ40×1	22.72	0.62
				Coil ⁽¹⁾ : 36.75 mT, 5Hz			30.1	0.82

#: This value is not provided directly in the original article but is obtained through our estimation.

*: This represents the maximum pumping flow rate when the pump body is operating. Since the article does not provide additional data, it cannot be converted into an average flow rate.

** : The article does not provide the specific size of the permanent magnet but does specify its motion period (T = 7 s). Based on this information, we estimated the equivalent driving cycle for comparison purposes.

Coil⁽¹⁾: For the energy-saving waveform with a frequency of 2Hz, a magnetic field of 50mT, and a duty cycle of 0.2 used in our work, the corresponding equivalent amplitude of a sinusoidal magnetic field with the same energy consumption is 36.75mT (**calculated using Eq. (9) in Text S2**).

TPR: Thermoplastic rubber materials. **NdFeB:** Neodymium-iron-boron. **PDMS:** polydimethylsiloxane.

Normalized shape change ratio: A dimensionless metric calculated by dividing the shape transformation amount by a characteristic dimension (e.g., diameter or length) and further normalizing it by the applied magnetic field strength.

Normalized flow rate: This is defined as the ratio of the average liquid pumping flow rate (except for Ref. 15, which reports the maximum flow rate) to the driving magnetic field strength.

Red font: This indicates that the driving magnetic field must be a gradient field, the field strength is relatively high (> 100 mT), the actuation frequency is low (< 1 Hz), the shape change ratio per unit magnetic field is limited ($< 6.8 \times 10^{-3} \text{ mT}^{-1}$), the flow rate is relatively low (< 22.72 mL/min), or the normalized flow rate is limited (< 0.62 mL/min/mT).

Comment 2: On page 3 line 72, “As a result, achieving large shape transformations under low magnetic fields remains largely unexplored, and the development of corresponding functional applications is still in its early stages.” How do you define large deformation? What is the standard for low magnetic field strength? It would be more convincing if you could give specific parameters.

Response 2: We sincerely thank you for your highly professional and constructive comments. Clarifying the definitions of “large deformation” and “low magnetic field strength” **has significantly enhanced the clarity of our research objectives and better highlighted the technical advantages of the developed actuator.** Your insightful suggestion has not only improved the logical flow and rigor of our manuscript but also enabled us to present specific parameters that more accurately demonstrate the actuator’s performance. This, in turn, strengthens the significance of our contribution in addressing key challenges within the field. We truly appreciate your thorough evaluation and valuable guidance.

In principle, both “large deformation” and “low magnetic field strength” are relative terms without universally strict thresholds. In particular, the definition of a low magnetic field can vary significantly depending on material properties, actuator design, and application scenarios. Different studies may adopt different ranges based on their performance requirements and experimental setups. In our revised manuscript, we offer working definitions for both terms **based on commonly reported parameter ranges in the literature and the practical challenges associated with achieving such parameter values (See newly added Text S1 and Table S1 in Supporting Information).**

- **Large deformation:** We define large deformation as a shape change ratio larger than 0.5, where the shape change ratio is calculated as the ratio of the actuator’s maximum deformation to a characteristic structural dimension (e.g., diameter or length). This threshold is meaningful because many existing magnetoactive soft actuators in the literature do not achieve this level. To support this definition, **we have added Table S1 in the Supporting Information** (with updated numbering), which compares the shape change performance of magnetoactive soft actuators under fully edge-constrained and zero pressure difference conditions.

- **Low magnetic field strength:** We define low magnetic field strength as $B < 100$ mT, which aligns with several prior works in the field where driving fields below 100 mT are explicitly described as “low” or “weak” or “small” (e.g., Refs. [1-3] in *Supporting Information*). Additionally, as shown in Table S1, a significant number of reported actuators operate under magnetic fields exceeding this threshold, further supporting the validity of our classification.

Notably, our actuator achieves a large shape change (ratio > 0.8) under a magnetic field strength below 50 mT, surpassing previously reported magnetoactive soft actuators (see Table S1).

We sincerely thank you again for your helpful comment. We believe that these clarifications and additions make the claims in our manuscript more scientifically grounded and convincing.

The newly added Text S1 in the *Supporting Information* is shown below.

Text S1: Definition of large deformation and low magnetic field strength in this work

In principle, both “large deformation” and “low magnetic field strength” are relative terms without universally strict thresholds. In particular, the definition of a low magnetic field can vary significantly depending on material properties, actuator design, and application scenarios. Different studies may adopt different ranges based on their performance requirements and experimental setups. In this work, we offer working definitions for both terms based on commonly reported parameter ranges in the literature and the practical challenges associated with achieving such parameter values (See Table S1).

- **Large deformation:** We define large deformation as a shape change ratio larger than 0.5, where the shape change ratio is calculated as the ratio of the actuator’s maximum deformation to a characteristic structural dimension (e.g., diameter or length). This threshold is meaningful because many existing magnetoactive soft actuators in the literature do not achieve this level, as shown in Table S1, which compares the shape change performance of magnetoactive soft actuators under fully edge-constrained and zero pressure difference conditions.

- **Low magnetic field strength:** We define low magnetic field strength as $B < 100$ mT, which aligns with several prior works in the field of magnetic actuation where driving fields below 100 mT are explicitly described as “low” or “weak” or “small”^[1-3]. Additionally, as shown in Table S1, a significant number of reported soft actuators operate under magnetic fields exceeding this threshold, further supporting the validity of our classification in the research scope of this work.

The newly added Table S1 in the Supporting Information is shown below.

Table S1. Comparison of shape transformation characteristics of magnetoactive soft actuators.*

Reference	Size (mm)	Shape change (mm)	Shape change ratio	Magnetic field (mT)	Normalized shape change ratio (mT ⁻¹)
[9]	Φ 20×0.5 Folded	5.6	0.280	40 mT	7×10^{-3}
	Φ 20×0.5 Flat	2.17	0.109	40 mT	2.71×10^{-3}
[10]	Φ 16×1.4	1.0	0.063	89.9 mT	7.01×10^{-4}
[11]	Φ 20×1	3.40 [#]	0.170	300 mT	5.67×10^{-4}
[12]	Φ 10×0.265	4.1	0.410	938 mT	4.37×10^{-4}
[13]	Φ 30×1.08	12.20 ^{##}	0.407	125 mT	3.25×10^{-3}
[14]	Φ 30×0.5	7.30	0.243	192 mT	1.27×10^{-3}
Our work	Φ 40×1	35.77	0.894	40 mT	2.24×10^{-2}
	Φ 20×0.5	16.22	0.811	20 mT	4.06×10^{-2}

*: To ensure a fair comparison, those soft actuators with fully edge-constrained configurations were considered in this table.

#: This value is not provided directly in the original article but is obtained through our estimation.

##: The article only reports the upward shape change. Based on the bistable symmetry of the spherical shell, we estimated the total shape change as twice the upward value.

Shape change ratio: A dimensionless value calculated by dividing the maximum deformation by a characteristic dimension (e.g., diameter or length).

Normalized shape change ratio: This is defined as the shape change ratio normalized by the applied magnetic field strength.

Red font: This indicates that the shape change ratio is less than 0.5, the driving magnetic field exceeds 100 mT, and the normalized shape change ratio is below 2.24×10^{-2} .

Comment 3: Are the currently designed driving magnetic fields (20 - 100 mT) close to the theoretical lower limit? Will the deformation speed and driving magnetic field strength be further reduced under smaller sizes and softer material systems? Is there a universal relationship between critical torque and shell stiffness/curvature?

Response 3: Thank you for raising these fundamental and insightful questions regarding the theoretical lower limit of the driving magnetic field, the effects of size and material properties, and the mechanical basis of the switching behavior. To maintain logical clarity, we respond in reverse order to match the structure of your questions:

(1) Is there a universal relationship between critical torque and shell stiffness/curvature?

For bistable spherical shells, a theoretical relationship may exist between the critical switching torque and the shell's bending stiffness ($D= Eh^3/12(1-\nu^2)$, E is the Young's modulus of the material, h is the shell thickness, and ν is the Poisson's ratio of the material) and curvature ($\kappa=1/R$, R is the shell radius), as these two parameters jointly govern the elastic energy landscape during deformation. According to classical shell theories (e.g., Koiter shell theory), a higher bending stiffness implies a stronger resistance to shape change. Thus, the critical magnetic torque—and correspondingly, the switching magnetic field—tends to increase with increasing stiffness, such as with stiffer materials (larger E) or thicker shells. This trend is supported by experimental data in our study (Fig. 2c), where shells made from stiffer materials exhibit significantly higher switching fields.

However, deriving a universal analytical expression for the critical torque remains extremely challenging due to several reasons: the inherent magneto-mechanical coupling, geometrical and material nonlinearities, and strong sensitivity to boundary conditions (such as edge constraints and field orientation). Under dynamic loading, additional complexities such as rate effects, geometric imperfections, and competing instability modes (e.g., global buckling vs. local

wrinkling) further obscure theoretical predictions. Therefore, while certain scaling laws and trends can be observed, a complete theoretical model remains an open topic for future work, as also discussed in our revised manuscript.

(2) Will the deformation speed and driving magnetic field strength be further reduced under smaller sizes and softer material systems?

Yes, the required driving magnetic field strength and the deformation speed of the actuator are indeed influenced by its size and material properties. As previously discussed, softer material systems—characterized by lower Young's modulus—exhibit reduced bending stiffness, which in turn lowers the magnetic torque needed to induce deformation. Consequently, the critical magnetic field required for actuation decreases. This trend is clearly illustrated in Fig. 2c. Additionally, softer materials generally respond more rapidly under magnetic stimuli—provided that viscoelastic damping (e.g., in highly dissipative rubbers) does not dominate the dynamics.

Regarding size effects, Fig. 2e shows that the switching magnetic field increases with thickness but decreases with shell radius. The increase in thickness significantly enhances the bending stiffness, thereby affecting both the deformation capability and the magnitude of the required switching magnetic field. In contrast, the influence of radius variation is quite complex. Although the bending stiffness is independent of the shell radius, notable changes occur in the shell volume, edge constraint conditions, and local strain distribution during deformation. Consequently, the observed increase in the switching magnetic field as the shell radius decreases arises from the combined effect of multiple interacting factors. These points have been further clarified and expanded upon in the revised manuscript to better explain the phenomenon (see **Section 2.1**).

(3) Are the currently designed driving magnetic fields (20 - 100 mT) close to the theoretical lower limit?

Not yet. The current driving field range (20-100 mT) does not represent the theoretical limit. There remains significant room for optimization by tuning

parameters such as material stiffness, shell geometry, magnetic field orientation, and spatial magnetization profile—all of which can affect the total magnetic torque. However, such optimization strategies lie beyond the scope of the present study. The primary goal of this work is to introduce a new type of magnetically actuated bistable shell actuator and to demonstrate its effectiveness and application potential in achieving large-scale shape transformation under low magnetic fields. It is worth noting that, even under the current design, our approach already surpasses existing methods in key performance metrics. Future research could explore targeted optimization strategies to further reduce the required driving field and enhance overall performance. These prospects have been discussed in the revised manuscript (**see Discussion Section**).

We sincerely thank you again for these insightful concerns. They have not only improved the rigor and clarity of the current work but also provided valuable inspiration for our future research.

In the first paragraph of Section 2.1, we made the following revisions:

It can be observed that the compression resistance of PDMS-based actuators is significantly higher than that of silicone-based actuators, exceeding 50 times, due to the higher Young's modulus of PDMS. This explains why, as shown in Fig. 2c(ii), although PDMS-based actuators exhibit bistable characteristics (states A and B), they struggle to switch states under the applied magnetic field. In contrast, silicone-based actuators can easily achieve state switching under the same magnetic conditions.

As shown in Fig. 2e, the switching magnetic field increases with shell thickness but decreases with shell radius. Similar to the effect induced by increasing the Young's modulus, a greater thickness substantially enhances the bending stiffness, thereby increasing the difficulty of deformation and the magnitude of the required switching magnetic field. In contrast, the mechanism underlying the influence of shell radius is more complex. Although the bending stiffness is theoretically independent of the shell radius, variations in radius lead to significant changes in shell volume, edge constraint

conditions, and local strain distribution during deformation. Consequently, the observed increase in the switching magnetic field with decreasing shell radius could result from the combined effect of these multiple interacting factors.

In the last paragraph of *Discussion Section*, we made the following revisions:

First, optimizing key parameters including shell geometry, material stiffness, and magnetic design (such as spatial magnetization profiles and field alignment) is essential for further reducing the magnetic field required for actuation. However, due to the inherent complexities of magneto-mechanical coupling, nonlinear large deformation, and sensitivity to boundary conditions, it remains challenging to accurately predict the critical switching torque or the required magnetic field. Therefore, developing theoretical models and accurate simulations that capture these effects will play an important role in guiding future actuator design.

Comment 4: Authors should carefully review their manuscripts before submission. On Page 10 line 296 - 307, the same content appears twice.

Response 4: Thank you for pointing out the duplication in lines 296 - 307 on Page 10. We have carefully reviewed the manuscript and confirmed that the content in these lines was mistakenly repeated. **The redundant part has now been removed in the revised manuscript.** We have conducted a thorough check of the entire manuscript to avoid similar issues.

Additional mark:

General Comment: The authors have fully addressed the concerns and comments raised by Reviewer 1 and have basically addressed those from Reviewer #2. However, some responses to Reviewer #2 would benefit from further revision.

Response: Thank you very much for your valuable feedback and positive evaluation of our previous revisions. We sincerely appreciate your recognition of our efforts in addressing the reviewers' comments. We will further improve our responses to Reviewer #2's comments as suggested, to ensure they are more comprehensive and clearly articulated.

Comment 5: Contribution to the field (response to Reviewer #2's overall comments): The authors' response regarding the study's contribution does not appear sufficiently convincing.

Response: We sincerely thank you for your valuable feedback and appreciate the opportunity to further clarify the contribution and novelty of our study. We acknowledge that our earlier response may not have clearly articulated the conceptual advances and technical distinctiveness of our work. **Building on the insightful suggestions provided during this review round, we have further refined our discussion of the study's innovation and implemented corresponding improvements throughout the manuscript.** In particular, we have substantially strengthened the *Abstract*, *Introduction* and *Discussion* sections to clearly highlight the unique aspects of our design strategy, scientific contributions, and application significance. Below, we provide a detailed clarification of how our study contributes to the field of magnetoactive soft actuators, distinguishing it from prior work in terms of design innovation, scientific insight, and application impact.

(1) Addressing a Critical Limitation in the Field of Magnetoactive Soft Actuators

Magnetoactive soft actuators are widely valued for their remote controllability, fast response, and potential for multifunctional integration. However, a persistent

challenge remains: **the requirement for continuous and relatively high magnetic fields (>100 mT) to maintain large-scale shape transformations (shape change ratio exceeding 0.5), particularly under constrained conditions.** This limitation significantly hampers their practical applicability, especially in scenarios that demand significant shape transformations and control.

Our work directly addresses this gap by proposing a magnetically actuated hemispherical bistable mechanism capable of reliable switching under low magnetic fields and strict boundary constraints, without the need for continuous excitation. This work achieves robust, reversible, and large-scale shape transformations with a shape change ratio exceeding 0.8 under low magnetic fields below 50 mT, significantly surpassing performance levels reported in the field of magnetoactive soft actuators.

(2) Demonstrated System-Level Applications with Functional Differentiation

To validate the versatility and practical value of our design, we implemented three proof-of-concept applications—a soft pump, a programmable metamaterial, and a variable-stiffness gripper—each leveraging distinct capabilities of the proposed actuator, beyond its shared feature of large, low-field deformation:

- The soft pump is driven by an energy-efficient sinusoidal waveform and achieves flow rates exceeding 30 mL/min under magnetic fields below 50 mT. It exhibits a normalized flow rate of 0.82 mL/(min·mT), significantly outperforming existing magnetoactive soft diaphragm pumps, and uniquely enables closed-loop flow control—a rare feature in such systems.
- The programmable metamaterial consists of discrete bistable units capable of encoding and reconfiguring spatial information, offering a structurally and operationally simple approach to building logic-embedded three-dimensional soft architectures.
- The biomimetic soft gripper achieves secure and adaptive grasping through magnetic stiffness control, showcasing a unique integration of sea anemone-inspired morphology with magnetically induced bistability.

These applications go beyond simple deformation demonstrations, collectively demonstrating how our actuator can serve as a core building block for multifunctional soft machines, with advantages in response speed, reconfigurability, design compactness, and tunable performance.

(3) Substantial Distinction from Existing Works

While magnetic bistability has been explored in previous studies, our work differs fundamentally in several key aspects, such as structural design, boundary conditions, and deformation mode. Specifically, the deformation mode enabled in our system—shell deformation under strict edge confinement—is distinct from prior designs such as the two-end-constrained beam structure by Pal & Sitti (PNAS, 2023) and the unconstrained pyramid architecture by Tang et al. (Sci. Robot., 2024). Moreover, unlike approaches that generate dome-shaped bistable structures through self-buckling (Chen et al., Adv. Funct. Mater., 2024) or externally applied pre-stress (Abbasi et al., Adv. Mater. Technol., 2024) from flat sheets, our method directly fabricates a three-dimensional hemispherical shell with embedded hard magnetic microparticles. This design intrinsically enhances spatial deformation during bistable transitions, enabling robust and large-scale shape transformations. Building upon this foundation, we further tailored the actuator’s structural configuration and the applied magnetic field design to meet the specific requirements of targeted application scenarios.

In summary, we believe this work makes a meaningful contribution to the field by addressing the challenge of achieving large-scale shape transformation under low magnetic fields through a structurally and magnetically integrated bistable actuation strategy. The corresponding supporting materials, clarifications, and additions have been detailed in the responses above and are reflected in the revised manuscript; thus, they are not repeated here for brevity.

Comment 6: For the response to Reviewer #2's comment 5 (Table S2). While the authors revised Table S2 in response to this comment, I remain concerned that the “snap time” column does not provide a fair comparison. The response time of the different bistable actuator is closely related to factors such as material stiffness and actuator size.

Response 6: We sincerely thank the reviewer for the continued attention to the fairness of comparisons in Table S3 (previously Table S2). We fully agree that snap time of bistable dome-shaped soft actuators depends on multiple parameters such as actuator size, material stiffness, and test conditions. To address this concern, we implemented the following targeted revisions:

- **Clarifying note: We added a statement in the note of Table S3:** “It should be noted that when comparing different studies, snap time should be considered a reference metric rather than a strict benchmark, as it is influenced by various factors such as actuator dimensions, material stiffness, and testing conditions. To improve data comparability, we tried to adopt comparable test conditions (e.g., similar actuator dimensions) in this table.”
- **Context for geometry differences: We added a new column listing the actuator dimensions (diameter and thickness) for each study.** This provides readers with direct contextual information when interpreting snap time values, enabling them to consider the potential influence of size on response speed and thus gain a clearer understanding of the factors underlying snap time differences between actuators.
- **Data verification and harmonization:** For the actuation parameter entry of Ref. [13] in *Supporting Information*, we replaced the previously listed 200 mT magnetic field value (derived from simulation data) with the correct experimental test value of 125 mT reported in the same paper. The actuator dimensions corresponding to this parameter fall within the size range of other studies in Table S3 (diameter 20–44 mm, thickness 0.3–3 mm), ensuring that our comparison is based on similar scale conditions.

- **Supplement of smaller-scale actuator data:** Considering the size range of actuators in the existing works listed in the table and the influence of actuator size on snap time, **we have supplemented the revised manuscript with experimental data on a magnetic bistable actuator with a diameter of 20 mm and thickness of 0.5 mm (see Table S3 and Movie S16).** This addition provides further reference points for parameter comparison.

We hope these clarifications improve the transparency of the comparison and fully address your concerns.

The revised Table S3 in the *Supporting Information* is shown below.

1 **Table S3. Comparison of bistable dome-shaped soft actuators.**

Reference	Actuation principle	Material type	Actuation parameter	Snap Time ³	Size (mm)	Application case
[19]	Pneumatic loading	Dragon Skin + Ecoflex 30 + Smooth-Sil 950	10.2 kPa and 3.3 kPa ¹	0.2 s	Φ 20×3	 ■ Soft gripper ■ Soft earthworm
[20]	Temperature	Hydrogel	From 20 °C to 40 °C (PH=7)	90 s	Φ 30×2.2	—
	PH		From PH = 2 to 7 (20 °C)	28 min		
[21]	Magnetic force	PDMS + Carbonyl iron particles	>200 mT ²	0.14 s	Φ 44×1.5	—
[13]	Magnetic torque and Pneumatic loading	Vinylpolysiloxane (VPS-32) + NdPrFeB particles	125 mT	—	Φ 30×1.08	 ■ Braille dot
[14]	Magnetic torque	PDMS + NdFeB particles	192 mT	0.12 s	Φ 30×0.5	 ■ Flytrap ■ Reconfigurable electronics ■ Dynamic bioreactor
Our work	Magnetic torque	Silicone Elastomer (Ecoflex00-20) + NdFeB particles	60 mT ($f = 6$ Hz)	67 ms	Φ 40×1	 ■ Soft Pump ■ Reprogrammable metamaterials ■ Bionic soft gripper
			40 mT ($f = 6$ Hz)	43 ms ⁴	Φ 20×0.5	

- 2 **1:** The two sets of pressures correspond to the two stable states of the spherical shell.
- 3 **2:** Our estimation is based on the reported limiting switching distance of 14 mm from the permanent magnet, together with the magnetic field strength distribution diagram provided in the study.
- 4 **3:** It should be noted that when comparing different studies, snap time should be considered a reference metric rather than a strict benchmark, as it is influenced by various factors such as
- 5 actuator dimensions, material stiffness, and testing conditions. To improve data comparability, we tried to adopt comparable test conditions (e.g., similar actuator dimensions) in this table.
- 6 **4:** This data is obtained from Supplementary Movie S16.
- 7 **Red font:** This indicates that these bistable actuators in the literature have a relatively long response time (>0.1 s) or require a relatively large magnetic field (> 100 mT).

Comment 7: Both Reviewers #1 and #2 mentioned issues with the rigor of the manuscript, especially typographical and grammatical errors (e.g., numbering, abbreviations, etc). Although the authors have made revisions, some lingering issues remain. For example, on Page 10 (line 296 - 307), the same content appears twice. The authors should carefully proofread the manuscript.

Response 7: We sincerely thank you, along with Reviewers #1 and #2, for your careful reading and valuable comments on the rigor and language quality of our manuscript. We deeply regret the oversight of these errors and apologize for any inconvenience caused. Upon receiving this feedback, **we conducted a thorough proofreading of the entire manuscript**, including typographical errors, repeated content, numbering inconsistencies, and abbreviation usage.